# Neutrophil extracellular trap formation and gene programs distinguish TST/IGRA sensitization outcomes among *Mycobacterium tuberculosis* exposed persons living with HIV

Elouise E. Kroon[1]*, Wilian Correa-Macedo[2,3,4], Rachel Evans[5,6], Allison Seeger[7], Lize Engelbrecht[8], Jurgen A. Kriel[8], Ben Loos[9], Naomi Okugbeni[1,10], Marianna Orlova[2,3,4], Pauline Cassart[2,3], Craig J. Kinnear[1,10], Gerard C. Tromp[1,11,12], Marlo Möller[1,11], Robert J. Wilkinson[7,13,14], Anna K. Coussens[5,6,7], Erwin Schurr[2,3,4], Eileen G. Hoal[1]

1 DSI-NRF Centre of Excellence for Biomedical Tuberculosis Research; South African Medical Research Council Centre for Tuberculosis Research; Division of Molecular Biology and Human Genetics, Faculty of Medicine and Health Sciences, Stellenbosch University, Cape Town, South Africa, 2 Program in Infectious Diseases and Immunity in Global Health, The Research Institute of the McGill University Health Centre, Montréal, Canada, 3 McGill International TB Centre, McGill University, Montréal, Canada, 4 Department of Biochemistry, McGill University, Montréal, Canada, 5 Infectious Diseases and Immune Defence Division, Walter and Eliza Hall Institute of Medical Research, Parkville, Australia, 6 Department Medical Biology (WEHI), Faculty of Medicine, Dentistry and Health Sciences, University of Melbourne, Parkville, Australia, 7 Wellcome Centre for Infectious Diseases Research in Africa, Institute of Infectious Disease and Molecular Medicine and Department of Medicine, University of Cape Town, Observatory, South Africa, 8 Central Analytical Facilities, Microscopy Unit, Stellenbosch University, Cape Town, South Africa, 9 Department of Physiological Sciences, Stellenbosch University, Stellenbosch, South Africa, 10 South African Medical Research Council Genomics Platform, Tygerberg, South Africa, 11 Centre for Bioinformatics and Computational Biology, University of Stellenbosch, Cape Town, South Africa, 12 SAMRC-SHIP South African Tuberculosis Bioinformatics Initiative (SATBBI), Center for Bioinformatics and Computational Biology, Cape Town, South Africa, 13 Department of Infectious Diseases, Imperial College London, London, United Kingdom, 14 The Francis Crick Institute, London, United Kingdom

* elouise_k@sun.ac.za

**Data Availability Statement:** Data from the manuscript is archived on the European Genome-

## Abstract

Persons living with HIV (PLWH) have an increased risk for tuberculosis (TB). After prolonged and repeated exposure, some PLWH never develop TB and show no evidence of immune sensitization to *Mycobacterium tuberculosis* (*Mtb*) as defined by persistently negative tuberculin skin tests (TST) and interferon gamma release assays (IGRA). This group has been identified and defined as HIV+ persistently TB, tuberculin and IGRA negative (HITTIN). To investigate potential innate mechanisms unique to individuals with the HITTIN phenotype we compared their neutrophil *Mtb* infection response to that of PLWH, with no TB history, but who test persistently IGRA positive, and tuberculin positive (HIT). Neutrophil samples from 17 HITTIN (PMN_HITTIN) and 11 HIT (PMN_HIT) were isolated and infected with *Mtb* H37Rv for 1h and 6h. RNA was extracted and used for RNAseq analysis. Since there was no significant differential transcriptional response at 1h between infected PMN_HITTIN and PMN_HIT, we focused on the 6h timepoint. When compared to uninfected PMN, PMN_HITTIN displayed 3106 significantly upregulated and 3548 significantly downregulated differentially expressed genes (DEGs) (absolute cutoff of a $\log_2$FC of 0.2, FDR < 0.05) whereas

phenome Archive (EGA) as "Neutrophils as effector cells in resistance to infection by Mycobacterium tuberculosis in HIV- infected individuals" with the study ID EGAS00001007262 (https://ega-archive.org/studies/EGAS00001007262). Qualified researchers can request access to the data by contacting the listed contact person on https://ega-archive.org/datasets/EGAD00001010893 and completing a Data Access Request form which will be evaluated by the Data Access Committee who will consult with the Health Research Ethics Committee of Stellenbosch and the Stellenbosch University Contracts Office. Data access will be for approved use and will be governed by the provisions laid out in the associated informed consent of individual data subjects and of the Health Research Ethics Committee of Stellenbosch University. All investigators will have to sign a Data Transfer Agreement as stipulated by the Stellenbosch University Contracts Office. A qualified researcher refers to a senior investigator who is employed or legitimately affiliated with an academic, non-profit or government institution and who has a track record in the field. Researchers granted access to Project Data must feedback the results of their research to the Data Access Committee, after publication for distribution to the community representative if required. Access is conditional upon availability of data and on signed agreement by the researcher(s) and the responsible employing institution to abide by the policies and conditions related to publication, data ownership, data return, intellectual property rights, data disposal, ethical approval, confidentiality and commercialization referred herein.

**Funding:** The work was funded by the South African Medical Research Council through its Division of Research Capacity Development under the SAMRC Clinician Researcher M.D PhD Development programme(EEK). The content of any Publications from any studies during this Degree are solely the responsibility of the authors and do not necessarily represent the official views of the South African Medical Research Council. This publication is also supported by NeutroTB which is part of the EDCTP2 programme supported by the European Union (grant number TMA2018CDF-2353-NeutroTB)(EEK). The views and opinions of authors expressed herein do not necessarily state or reflect those of EDCTP. In addition, the work is part of an overall project funded by National Institutes of Health (NIH 1R01AI124349)(ES, EGH, MM) and was also funded by the Walter and Eliza Hall Institute of Medical Research(AKC). The funders had no role in study design, data collection and analysis, decision to publish, or preparation of the manuscript.

$PMN_{HIT}$ demonstrated 3816 significantly upregulated and 3794 significantly downregulated DEGs following 6h *Mtb* infection. Contrasting the $log_2FC$ 6h infection response to *Mtb* from $PMN_{HITTIN}$ against $PMN_{HIT}$, 2285 genes showed significant differential response between the two groups. Overall $PMN_{HITTIN}$ had a lower fold change response to *Mtb* infection compared to $PMN_{HIT}$. According to pathway enrichment, Apoptosis and NETosis were differentially regulated between HITTIN and HIT PMN responses after 6h *Mtb* infection. To corroborate the blunted NETosis transcriptional response measured among HITTIN, fluorescence microscopy revealed relatively lower neutrophil extracellular trap formation and cell loss in $PMN_{HITTIN}$ compared to $PMN_{HIT}$, showing that $PMN_{HITTIN}$ have a distinct response to *Mtb*.

## Author summary

Neutrophils can contribute to the severity of tuberculosis (TB). More recently TB protective mechanisms of neutrophils have also been highlighted. We examined the gene expression of neutrophils, one of the first immune cells to make contact with *Mycobacterium tuberculosis* (*Mtb*), the causative organism of TB, once *Mtb* is inhaled. Our study used two populations: i) persons who are living with HIV (PLWH), have no history of TB and test persistently negative for *Mtb* T cell sensitization (HITTIN); and ii) PLWH who have a positive *Mtb*-specific T cell response with no previous TB (HIT). Specifically, we compared neutrophil response to *Mtb* and compared HITTIN to HIT. Compared to neutrophils from HIT persons, HITTIN neutrophils ($PMN_{HITTIN}$) show significant gene expression differences in response to *Mtb* infection. Unexpectedly, our results show $PMN_{HITTIN}$ have a global lower response to *Mtb* infection with fluorescent microscopy revealing lower neutrophil extracellular trap (NET) formation in $PMN_{HITTIN}$ compared to $PMN_{HIT}$ after 6h infection with *Mtb*. NETs are weblike structures and are formed from decondensed DNA during a unique form of cell death named NETosis. This data highlights $PMN_{HITTIN}$ as important immune cells in response to *Mtb* in HITTIN.

## Introduction

Worldwide, tuberculosis (TB) remains the leading cause of death by a single bacterial agent [1]. People who are living with HIV (PLWH) have an increased risk of TB [2,3]. We previously identified a cohort of PLWH, living in a community with high TB burden in the Western Cape, South Africa, who are persistently TB, tuberculin skin test (TST) and interferon gamma release immune assay (IGRA) negative (HITTIN) [4–6]. TST and IGRA measure persistent adaptive immune memory to *Mycobacterium tuberculosis* (*Mtb*) protein antigens and are used as a surrogate for prior *Mtb* infection. Despite these persons having a history of previously low CD4+ T-cell counts, prior to antiretroviral therapy (ART), they display no canonical evidence of prior *Mtb* infection or history of disease [4]. However, we did detect circulating *Mtb*-specific antibodies, despite absence of canonical T-cell memory to dominant *Mtb* antigens, suggesting these individuals may possess a unique immune mechanism of TB protection, not reliant on conventional T-cell help.

We hypothesized that the innate immune system, and specifically neutrophils (PMN), play an inherent role in the protective control of *Mtb* infection in HITTIN. PMN are the most

**Competing interests:** I have read the journal's policy and the authors of this manuscript have the following competing interests: RJW reports support from the Francis Crick Institute which receives its core funding from Cancer Research UK (CC2112), the UK Medical Research Council (CC2112), and the Wellcome Trust (CC2112); and grants from National Institutes of Health, during the conduct of the study. AKC reports grants from South African Medical Research Council (SHIP-02-2013), National Institutes of Health (U19AI111276), DFID/MRC/NIHR/Wellcome (MR/V00476X/), Australian Respiratory Council and Walter and Eliza Hall Institute of Medical Research during the conduct of the study. MM and EGH reports grants from National Institutes of Health (NIH 1R01AI124349), during the conduct of the study. ES reports grants from NIH 1R01AI124349 and the Canadian Institutes of Health Research (CIHR) through grant FDN-143332 for which ES is the PI. EEK reports grants from National Institutes of Health (NIH 1R01AI124349), other from European and Developing Countries Clinical Trials Partnership (grant number TMA2018CDF-2353-NeutroTB), other from South African Medical Research Council during the conduct of the study. GCT, BL, CJK, LE, JAK, AS, WCM, RE, PC, NO and MO report no competing interests.

abundant leukocytes and among the first responders to *Mtb* infection in the lung in animal models as well as humans [7]. They are armed with an arsenal of antimicrobial granules known to restrict *Mtb* growth and are key players in the inflammatory response against *Mtb* [8–11]. PMN can control *Mtb* growth during acute infection [8,11,12]. Household pulmonary TB contacts with higher initial peripheral neutrophil counts were less likely to become infected with *Mtb* [11]. Despite lower RNA expression in PMN compared to other innate immune cells, pathogen-triggered gene expression changes underlie microbial responses by PMN [13,14].

The mechanism of cell death can influence the outcome of *Mtb* infection control. Neutrophils control inflammation, predominantly through apoptotic and cell clearance mechanisms with efferocytosis of *Mtb*-infected apoptotic neutrophils by macrophages favoring a beneficial host outcome [14–18]. Neutrophil extracellular traps (NETs) were recently highlighted as a potentially important mechanism of resistance to TST/IGRA conversion in a Ugandan cohort [19]. As a primary cell death effector mechanism, neutrophils decondense their chromatin following histone modification, unraveling their DNA, which is then coated in cytoplasmic granular proteins before the DNA/histone/protein complex is expelled from the lysing neutrophil. This released web-like structured NET is able to capture extracellular microbes and kill them using the attached granule-derived antimicrobial peptides and histones [20,21]. *Mtb*-induced NETs are phagocytosis and reactive oxidative species (ROS) dependent, but have been suggested to lack *Mtb* microbicidal activity [22,23]. Recent studies point to potential crosstalk of NETs and forms of necrotic cell death pathways such as pyroptosis, despite the classical definition of NETosis as a unique mechanism of cell death [24]. Necrotic neutrophil cell death, uncontrolled type 1 interferon (IFN) responses and abundant ROS drive a hyperinflammatory neutrophil phenotype and contribute to TB severity [18,25–28].

The aim of this study was to gain insight into mechanisms underlying early neutrophil responses to *Mtb* in HITTIN persons with apparent protection from *Mtb* infection as inferred from the absence of canonical T-cell memory. Specifically, we tested if neutrophils from HITTIN study participants (PMN$_{HITTIN}$) displayed a transcriptional response to *Mtb* infection that was significantly different from that found in neutrophils from PLWH who have a robust *Mtb*-specific T-cell response, testing persistently TST and IGRA positive, but also with no history of TB (HIT). We demonstrated that the *in vitro Mtb*-induced gene expression changes in neutrophils from the HITTIN and HIT participants were significantly different. Genes differentially regulated by *Mtb* between the two groups were enriched for key regulators of ROS and NET formation. Fluorescence microscopy images corroborated the transcriptomic findings with PMN$_{HIT}$ demonstrating a greater response to *Mtb* with more NETs forming after infection. PMN$_{HITTIN}$ responded with less NETs and lower counts of key genes associated with nicotinamide adenine dinucleotide phosphate hydrogen (NADPH) activity and ROS formation possibly contributing to greater effector control of *Mtb*.

# Results

## Study participants and samples

Individuals included in the HITTIN and HIT groups were a part of the large ResisTB cohort [4]. Neutrophils obtained from 17 HITTIN and 11 HIT individuals, all of whom were PLWH and on ART, were used in the final analysis (Table 1). These individuals were part of stringently defined cohorts living in a high TB burden community who, despite low CD4+counts before ART initiation, never developed TB.

The average age of participants in the HITTIN group was 43.71 (±6.43) years and that of HIT was 44.09 (±6.89) (Table 1). There were no significant differences between the ratio of

**Table 1. Participant characteristics.**

| Characteristic | HITTIN[a], N = 17* | HIT[b], N = 11* | p-value** |
|---|---|---|---|
| **Age** | 43.71 (6.43) | 44.09 (6.89) | 0.9 |
| **Sex** | | | 0.6 |
| Female | 13 / 17 (76%) | 10 / 11 (91%) | |
| Male | 4 / 17 (24%) | 1 / 11 (9.1%) | |
| **Weight** | 75.81 (17.26) | 84.09 (18.12) | 0.3 |
| (Missing) | 1 | 0 | |
| **Height** | 1.63 (0.09) | 1.67 (0.06) | 0.2 |
| **BMI[c]** | 28.87 (7.38) | 30.15 (5.70) | 0.3 |
| (Missing) | 1 | 0 | |
| **Time on ART[d]** | | | 0.2 |
| Average time in years | 7.35 (3.15) | 8.97 3.2) | |
| **CD4 prior to starting ART[d]** | | | 0.4 |
| < 200 CD4+ cells/mm3 | 17 / 17 (100%) | 10 / 11 (91%) | |
| Between 200–350 CD4+ cells/mm3 | 0 / 17 (0%) | 1 / 11 (9.1%) | |
| **Last CD4 prior to enrolment** | 557.76 (229.70) | 486.73 (236.91) | 0.5 |
| **Last VL[e]** | | | 0.9 |
| $\leq$100 | 4/ 17 (24%) | 4 / 11 (36%) | |
| 124 | 1 / 17 (5.9%) | 0 / 11 (0%) | |
| LDL[f] | 12 / 17 (71%) | 7 / 11 (64%) | |
| **Current IPT[g]** | 1 / 17 (5.9%) | 0 / 11 (0%) | >0.9 |
| **Previous IPT treatment as per months of treatment** | | | 0.2 |
| 0 | 2 / 17 (12%) | 3 / 11 (27%) | |
| 2 | 0 / 17 (0%) | 1 / 11 (9.1%) | |
| 3 | 0 / 17 (0%) | 0 / 11 (0%) | |
| 4 | 0 / 17 (0%) | 0 / 11 (0%) | |
| 6 | 0 / 17 (0%) | 2 / 11 (18%) | |
| 12 | 10 / 17 (59%) | 5 / 11 (45%) | |
| 24 | 1 / 17 (5.9%) | 0 / 11 (0%) | |
| 36 | 3 / 17 (18%) | 0 / 11 (0%) | |
| Current | 1 / 17 (5.9%) | 0 / 11 (0%) | |
| **Chronic Illness** | 2 / 17 (12%) | 3 / 11 (27%) | 0.4 |
| **Alcohol Use** | 7 / 17 (41%) | 2 / 11 (18%) | 0.2 |
| **Smoker** | 2 / 17 (12%) | 0 / 11 (0%) | 0.5 |
| **Recreational Substance (Cannabis) use** | 0 / 17 (0%) | 1 / 11 (9.1%) | 0.4 |

*Mean (SD); n / N (%)

**Wilcoxon rank sum test; Fisher's exact test

[a] HITTIN (HIV-1-infected persistently TB, tuberculin and IGRA negative)

[b] HIT (HIV-1-infected IGRA positive tuberculin positive)

[c] BMI (body mass index) was calculated as weight (kg)/height$^2$(m)

[d] ART (antiretroviral therapy)

[e] VL (viral load)

[f] LDL (lower than detectable level)

[g] IPT (Isoniazid preventive therapy)

females to males in each group (p = 0.6, Fisher's exact test), age (p = 0.9, Wilcoxon rank sum test), or BMI (p = 0.6, Wilcoxon rank sum test) of the participants. All but one participant had CD4+ counts of < 200 cells/mm$^3$ prior to starting ART. Participants had controlled viral

loads, with participants in the HITTIN group having been on ART for 7.35 (±3.15) and HIT for 8.97 (±3.2) years. Although HITTIN was on ART for a shorter period there was no significant difference to the time spent in ART in HIT (p = 0.2, Wilcoxon rank sum exact test). One participant was on isoniazid *Mtb* prophylactic therapy (IPT) at the time of the enrolment, this was corrected for during the analysis. All PLWH receive IPT according to national guidelines in South Africa. There was no significant difference between the groups in terms of having taken prior IPT and time spent on IPT (p = 0.2, Fisher's exact test). HITTIN and HIT have a similar distribution of chronic disease (p = 0.4, Fisher's exact test), and social habits such as cigarette smoking (p = 0.5, Fisher's exact test), alcohol use (p = 0.2, Fisher's exact test) and cannabis use (p = 0.4, Fisher's exact test).

## Gene expression analysis of PMN in HITTIN and HIT in response to *Mtb*

Neutrophils were isolated from whole blood and infected with *Mtb* H37Rv for 1h and 6h. Purity of isolated neutrophils was confirmed by flow cytometry, with similar proportions of CD45+CD15+CD66b+CD16+ neutrophils isolated for HITTIN and HIT individuals (p = 0.28, Wilcoxon rank sum test) (S1 and S2 Tables). RNA was extracted from uninfected and infected neutrophils (PMN) at 1h and 6h timepoints and differential gene expression analyses performed by RNAseq. Significant differentially expressed genes (DEGs) were defined by an absolute cutoff of a $\log_2$FC of 0.2 and a false discovery rate (FDR) adjusted p-value of 5% (Tables 2 and S3).

*Mtb* infection triggered significant gene expression changes when compared to uninfected PMN at 1h: 151 up- and 40 downregulated genes for PMN$_{HITTIN}$ while 98 genes were up- and 11 were downregulated for PMN from the HIT group (PMN$_{HIT}$) (S1A and S1B Fig, S3 Table). A higher number of *Mtb* activated gene expression changes were observed at 6h post infection compared to 6h uninfected with 3106 up- and 3548 downregulated DEGs for HITTIN and 3816 up- and 3794 downregulated DEGs for HIT (Fig 1A and 1B, S3 Table).

**Table 2. Number of differentially expressed genes and their pathway and GO term enrichment for PMN$_{HITTIN}$ and PMN$_{HIT}$ in response to *Mtb*.**

| Contrast | DEGs[a] | # Genes FDR[b] ≤ 0.05 | # combined terms | KEGG FDR≤ 0.1 | Reactome FDR ≤ 0.1 | GO BP[c] FDR ≤ 0.1 |
|---|---|---|---|---|---|---|
| **PMN$_{HIT}$** [d] **6 hours(h) infected vs uninfected** | Total | 7610 | 701 | 65 | 15 | 621 |
| | Upregulated | 3816 | 915 | 53 | 51 | 811 |
| | Downregulated | 3794 | 10 | 0 | 10 | 0 |
| **PMN$_{HITTIN}$** [e] **6h infected vs uninfected** | Total | 6654 | 498 | 51 | 16 | 431 |
| | Upregulated | 3106 | 769 | 54 | 30 | 685 |
| | Downregulated | 3548 | 7 | 0 | 6 | 1 |
| **PMN$_{HITTIN}$—PMN$_{HIT}$ 6h infected**[f] | Total[g] | 2285 | 496 | 111 | 86 | 299 |
| | Less Upregulated[h] | 1217 | 719 | 48 | 55 | 616 |
| | Less Downregulated[i] | 1068 | 29 | 13 | 12 | 4 |

[a] DEGs (Differentially expressed genes)

[b] FDR (false discovery rate)

[c] GO BP (Gene Ontology Biological Processes)

[d] PMN$_{HIT}$ (HIV-1-infected IGRA positive tuberculin positive)

[e] PMN$_{HITTIN}$ (neutrophils from HIV-1-infected persistently TB, tuberculin and IGRA negative)

[f] Refers to the interaction test looking at the difference in response to *Mtb* infection after 6h between PMN$_{HITTIN}$ and PMN$_{HIT}$

[g] Refers to combined responses with significant positive and negative $\log_2$ fold change ($\log_2$FC) from the interaction test

[h, i] Refers to the negative[h] and positive[i] $\log_2$FC results from the interaction test

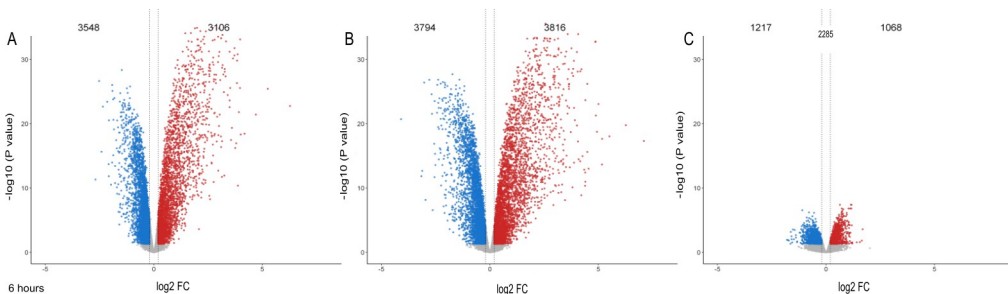

**Fig 1. Volcano plots of differential gene expression at 6h *Mtb* infection by PMN from HITTIN and HIT.** Volcano plot for transcriptional responses to *Mtb* challenge for neutrophils from HITTIN (PMN$_{HITTIN}$) and HIT (PMN$_{HIT}$) participants at 6h (**A-C**) post *Mtb* infection. The y-axis shows the negative log10 unadjusted P value and the x-axis the log$_2$ fold change (FC). The vertical dashed lines represent log$_2$FC thresholds of -0.2 and 0.2. Each gene is represented by a dot. Genes which are downregulated or upregulated as determined by the FDR ≤ 5% are shown in blue and red, respectively. Genes with non-significant expression changes and below the log 2FC threshold are shown in grey. Differentially expressed genes at 6h post-infection compared to 6h uninfected PMN from HITTIN (PMN$_{HITTIN}$) (**A**) and HIT participants (PMN$_{HIT}$) (**B**). Significant differentially triggered genes between PMN$_{HITTIN}$ and PMN$_{HIT}$ at 6h post infection (**C**).

When comparing transcriptional responses of infected PMN$_{HITTIN}$ and PMN$_{HIT}$ directly, no significant DEG were identified at 1h (S1C Fig, S3 Table). However, contrasting the effect differences for *Mtb* infection compared to no infection at 6h for each group (PMN$_{HITTIN}$ vs PMN$_{HIT}$), we identified 2285 genes with significant differential response between the two groups (Fig 1C, S3 Table). Since the 1h time point showed limited differences in gene induction by *Mtb*, we focused on the 6h time point differences between PMN phenotypes in subsequent analyses.

## Differential gene expression analysis showed a lower overall fold change difference during *Mtb* infection of PMN from HITTIN compared to HIT

We next compared the overall difference in transcriptomic response to *Mtb* between the two PMN groups. Consistent with different numbers of DEGs identified for each phenotypic group, when we evaluated the statistical significance of expression changes between the HITTIN and HIT groups, we identified an overall dampened 6h transcriptomic response in PMN$_{HITTIN}$ compared to PMN$_{HIT}$ ($p < 2.2e^{16}$) (Fig 2). Despite overall lower log$_2$FC among HITTIN, there was a strong correlation of log$_2$FC values for each gene between PMN$_{HITTIN}$ and PMN$_{HIT}$, suggesting that *Mtb* responses globally are conserved across phenotypes (S2 Fig). Irrespective of up- or downregulation of specific genes, the absolute response to *Mtb* after 6h was always smaller in PMN$_{HITTIN}$.

## Pathway and gene ontology (GO) enrichment analysis for DEGs between PMN$_{HITTIN}$ and PMN$_{HIT}$ after 6h *Mtb* infection

We next investigated the 2285 genes with significant differential transcriptional response of PMN$_{HITTIN}$ and PMN$_{HIT}$ to *Mtb* after 6 hours infection by examining the different pathways and analyzing the biological processes that are characterized by these genes. We conducted separate term enrichment analyses of up- and downregulated genes and a third analysis that considered both up- and downregulated genes. GO terms and Kyoto Encyclopedia of Genes and Genomes (KEGG) and Reactome pathways were considered significant if a term was enriched for at least 5 DEGs and a FDR cutoff of 10%. An overview of the results is shown in

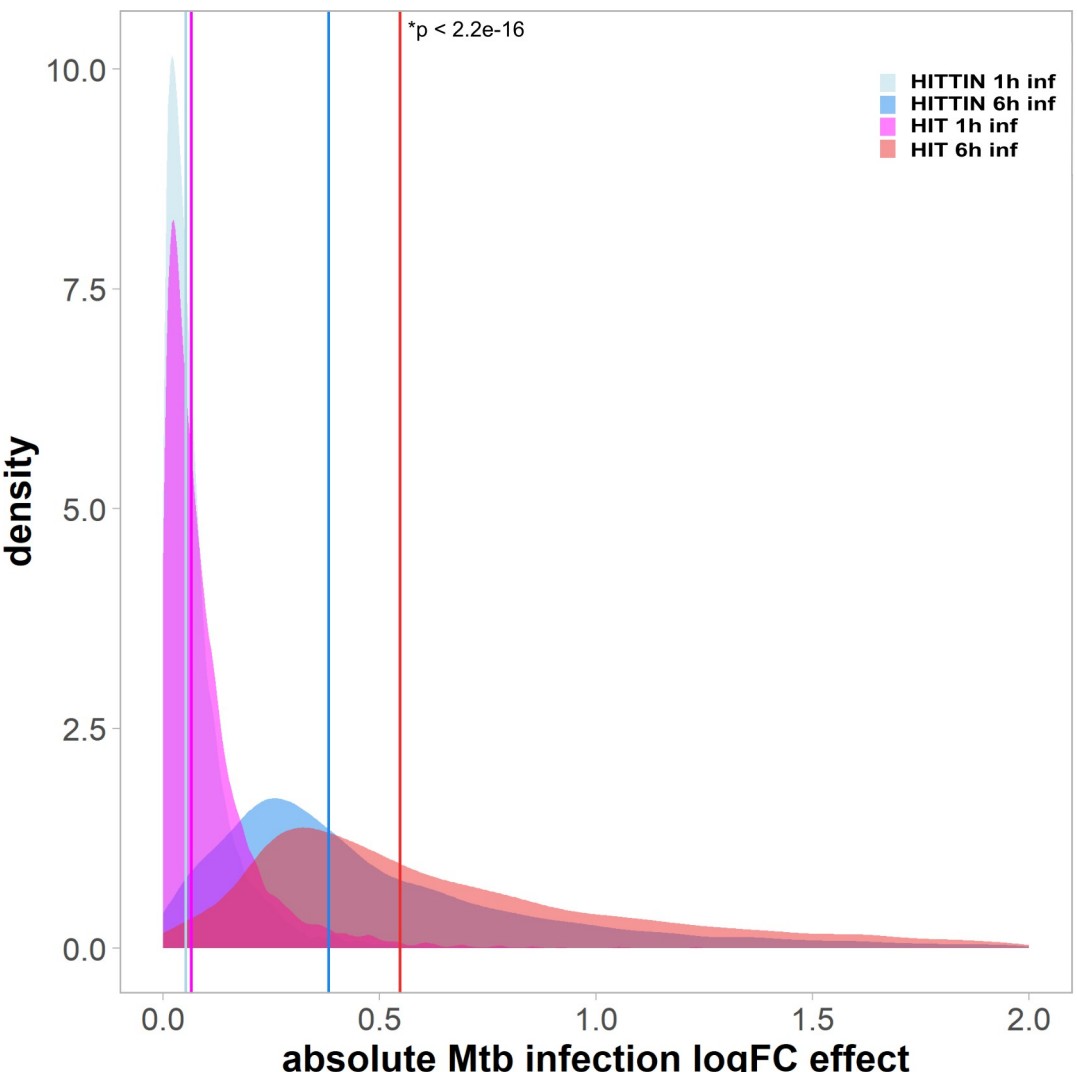

**Fig 2. The absolute log fold change *Mtb* infection effect at 1 and 6 h for PMN$_{HITTIN}$ and PMN$_{HIT}$.** The density plot shows the absolute log fold change (logFC) *Mtb* infection effect of each group at 1h and 6h. All DEGs from Fig 1 had their logFC converted to absolute values and plotted using density function. Absolute values for neutrophil DEGs from HITTIN participants are shown in light blue at 1h (median = 0.051) and a darker blue at 6h (median = 0.383). Absolute values for neutrophil DEGs from HIT participants are shown in magenta at 1h (median = 0.065) and red at 6h (median = 0.550). Vertical coloured lines indicate the median of absolute values. PMN$_{HITTIN}$ show a significantly lower logFC response to *Mtb* infection at 6h (p < 2.2e-16, Wilcoxon rank sum test) compared to 1h of infection with *Mtb* (p = 0.8255, Wilcoxon rank sum test).

Table 2. By focusing on all DEGs significantly different between PMN$_{HITTIN}$ and PMN$_{HIT}$, we observed a total of 496 enriched KEGG, Reactome and GO terms (Table 2). When evaluating genes more strongly triggered in PMN$_{HITTIN}$ we detected 29 terms. Conversely, when focusing on genes less strongly triggered in PMN$_{HITTIN}$ and comparatively to PMN$_{HIT}$, we detected 719 terms (Table 2).

Manhattan plots for the three term analyses at 6h are shown in Fig 3 with significantly different terms and pathways of interest indicated. Amongst the enriched terms in PMN$_{HITTIN}$, were "Apoptosis", "Neutrophil extracellular trap formation", and "NADPH regeneration", which are terms that directly relate to microbicidal activity of PMN$_{HITTIN}$ (Fig 3B). Although

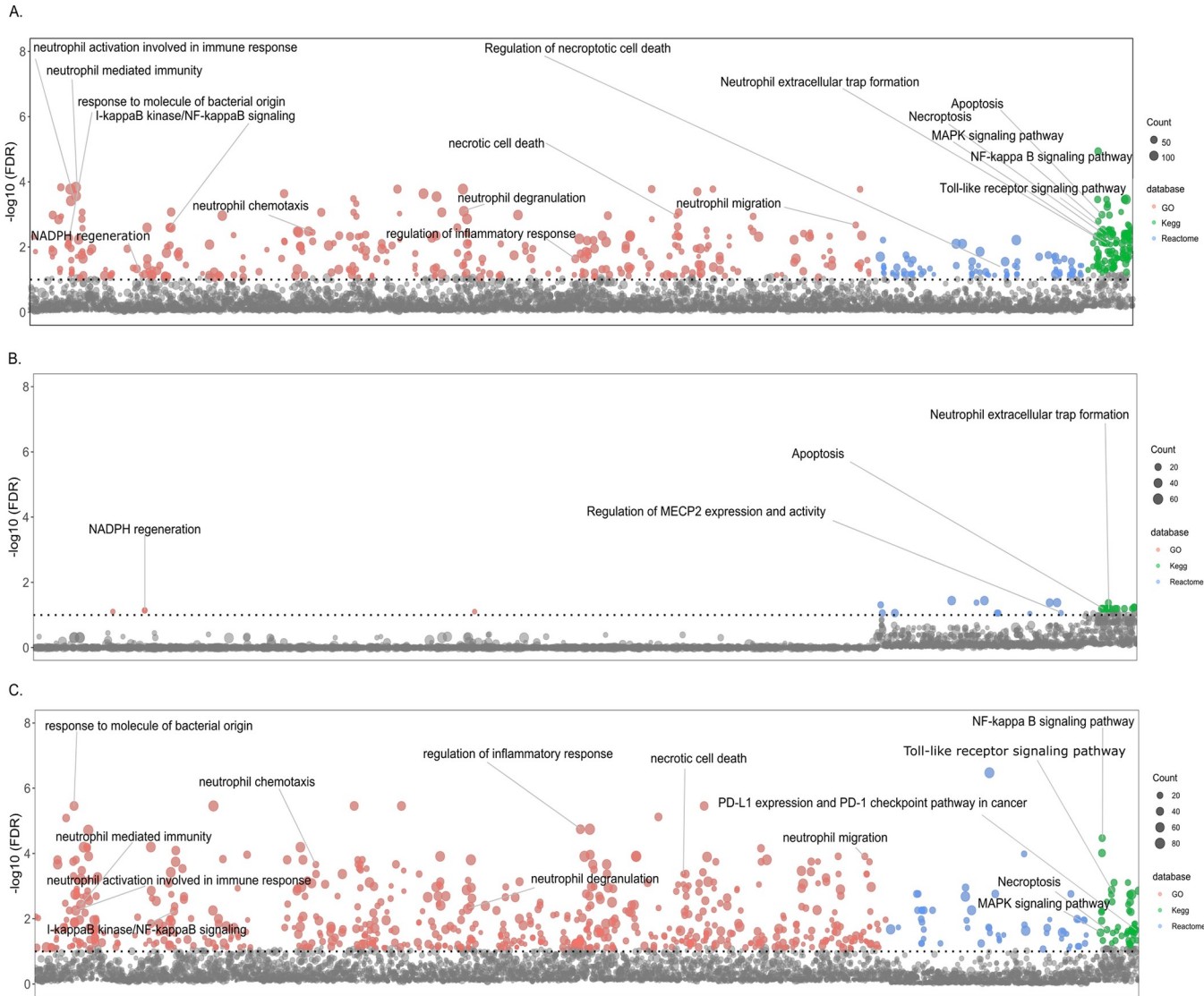

**Fig 3. Manhattan plot for enrichment tests of GO terms and Kegg and Reactome pathways.** The Manhattan plot shows pathways and GO terms for the DEGs triggered significantly differentially by *Mtb* in neutrophils from HITTIN versus neutrophils from HIT participants after 6h of infection. The tested terms are distributed along the x-axis. The y-axis represents the negative log 10 false discovery rate (FDR) result with the horizontal dotted line indicating the 10% FDR significance cut-off. Grey dots represent terms not meeting the significance threshold. Coloured dots represent significant terms from GO (red), Kegg (green) and Reactome (blue). The number of DEGs in each term is represented by the scaled size of the dot. Panel (**A**) shows the terms detected by all significantly different DEGs in the HITTIN vs HIT contrast. Panel (**B**) shows significant terms which are driven by genes with significant positive FC/less downregulated by PMN from HITTIN participants while panel (**C**) displays terms for genes significantly less upregulated for PMN from HITTIN participants as compared to PMN_HIT after 6h of *Mtb* infection.

these terms are mostly driven by genes with significant positive fold change (less downregulated) (Fig 3B, Table 3), the overall enrichment of the terms is also influenced by genes which are less up- and downregulated (Fig 3A, Table 3). This was also the case for terms with genes triggered relatively less strongly in PMN_HITTIN compared to PMN_HIT. These were dominated by genes involved in neutrophil chemotaxis, neutrophil degranulation, necroptosis and necrotic death (Fig 3C). It is important to note that the overall response to *Mtb* was always lower in PMN_HITTIN compared to PMN_HIT.

**Table 3. Top significantly differentially expressed up- and downregulated genes of interest in the "Neutrophil Extracellular Trap Formation" KEGG pathway, at 6hr of *Mtb* infection between PMN$_{HITTIN}$ and PMN$_{HIT}$.**

| Ensembl gene ID | Gene | Log$_2$FC after 6h *Mtb* infection | | | |
|---|---|---|---|---|---|
| | | PMN$_{HITTIN}$[a] | PMN$_{HIT}$[b] | PMN$_{HITTIN}$ x PMN$_{HIT}$[c] | adj.P.Val[d] |
| ENSG00000125730 | C3 | 2,167 | 3,0342 | -0,8672 | 7,24E-05 |
| ENSG00000172232 | AZU1 | -0,2334 | 0,4615 | -0,6949 | 0,0457 |
| ENSG00000103569 | AQP9 | 1,4303 | 1,9949 | -0,5646 | 5,15E-05 |
| ENSG00000165168 | CYBB | 0,603 | 1,0804 | -0,4775 | 0,0003 |
| ENSG00000109320 | NFKB1 | 2,1359 | 2,6025 | -0,4666 | 0,0089 |
| ENSG00000169032 | MAP2K1 | 0,5357 | 0,9572 | -0,4216 | 0,0004 |
| ENSG00000196954 | CASP4 | 0,539 | 0,96 | -0,421 | 0,0004 |
| ENSG00000137462 | TLR2 | 0,3467 | 0,7557 | -0,409 | 0,0154 |
| ENSG00000136869 | TLR4 | 0,8416 | 1,2239 | -0,3823 | 0,006 |
| ENSG00000137752 | CASP1 | 1,028 | 1,4004 | -0,3724 | 0,0085 |
| ENSG00000145675 | PIK3R1 | -0,1129 | 0,253 | -0,3659 | 0,0182 |
| ENSG00000197548 | ATG7 | 1,4527 | 1,808 | -0,3553 | 0,0094 |
| ENSG00000051523 | CYBA | 0,9853 | 1,3244 | -0,3391 | 0,0196 |
| ENSG00000128340 | RAC2 | 0,3795 | 0,6811 | -0,3015 | 0,0012 |
| ENSG00000116701 | NCF2 | -0,1042 | 0,1583 | -0,2625 | 0,0073 |
| ENSG00000197405 | C5AR1 | -0,2218 | -0,0056 | -0,2162 | 0,0258 |
| ENSG00000105221 | AKT2 | -0,1992 | -0,4116 | 0,2124 | 0,0123 |
| ENSG00000075624 | ACTB | -0,3289 | -0,5483 | 0,2195 | 0,045 |
| ENSG00000126934 | MAP2K2 | -0,2818 | -0,5169 | 0,2351 | 0,0304 |
| ENSG00000171608 | PIK3CD | -0,0918 | -0,3555 | 0,2637 | 0,0151 |
| ENSG00000005844 | ITGAL | -0,0544 | -0,3497 | 0,2952 | 0,0005 |
| ENSG00000113648 | MACROH2A1 | -1,2462 | -1,5613 | 0,3151 | 0,0234 |
| ENSG00000197943 | PLCG2 | -0,5073 | -0,8366 | 0,3293 | 0,0007 |
| ENSG00000166501 | PRKCB | -0,4236 | -0,758 | 0,3343 | 0,0005 |
| ENSG00000116478 | HDAC1 | -0,1265 | -0,505 | 0,3785 | 0,0011 |
| ENSG00000159339 | PADI4 | -0,2054 | -0,5956 | 0,3902 | 0,0007 |
| ENSG00000171720 | HDAC3 | -0,4041 | -0,81 | 0,4059 | 0,0004 |
| ENSG00000100030 | MAPK1 | -0,6563 | -1,0774 | 0,4211 | 0,0004 |
| ENSG00000164032 | H2AZ1 | -0,2065 | -0,6703 | 0,4638 | 0,0002 |
| ENSG00000246705 | H2AJ | 0,0745 | -0,3918 | 0,4663 | 0,0078 |
| ENSG00000102882 | MAPK3 | -0,7807 | -1,2548 | 0,474 | 0,0009 |
| ENSG00000110876 | SELPLG | -1,5235 | -2,0183 | 0,4948 | 0,0153 |
| ENSG00000068024 | HDAC4 | -0,3348 | -0,8415 | 0,5067 | 5,05E-05 |
| ENSG00000142208 | AKT1 | -0,4288 | -0,9999 | 0,5711 | 3,15E-05 |
| ENSG00000197837 | H4-16 | -0,0671 | -0,6704 | 0,6033 | 0,0073 |
| ENSG00000104518 | GSDMD | -0,2606 | -0,9556 | 0,6951 | 0,0004 |
| ENSG00000277224 | H2BC7 | -0,6019 | -1,6061 | 1,0041 | 0,0252 |

[a] The average log$_2$FC *Mtb* infection compared to the uninfected response for neutrophils from HITTIN at 6h.

[b] The average log$_2$FC *Mtb* infection compared to the uninfected response for neutrophils from HIT at 6h.

[c] The average log$_2$FC response difference between neutrophils from HITTIN and HIT in response to *Mtb* infection at 6h (interaction test).

[d] The adjusted p-value after the Benjamini Hochberg correction for multiple testing for interaction test looking at the differential response of HITTIN to HIT at 6h infected ("**PMN$_{HITTIN}$ x PMN$_{HIT}$**[*]") Significant genes were defined as genes with an absolute log$_2$FC $\geq$ 0.2 and adjusted p value $\leq$ 0.05

## NET area change difference between HITTIN and HIT from 1 to 6h after *Mtb* infection

To directly evaluate the biological outcome in the total amount of NETs observed between HITTIN vs HIT, as is suggested by our transcriptional enrichments, we used fluorescence microscopy to quantitate NET formation.

We stained fixed cells which were processed in parallel with the cells used for RNAseq. NETs were stained using anti-H2AH2B/DNA (PL2-3) which detects decondensed chromatin and nuclear DNA stained with Hoechst, and the area of both features was quantified.

As a measure of cellular viability over time we compared the total change in cell nuclei area between cells fixed at 1h and 6h of infection with *Mtb*. There was no significant two way interaction between $PMN_{HITTIN}$ and $PMN_{HIT}$ and the infection status, $F(1,12) = 1.1870$, $p = 0.30$, including a similar trend in direction of response for both PMN. However, there was a significant *Mtb* infection effect, $F(1,12) = 9.7290$, $p = 0.009$, with pairwise comparisons showing a significantly greater decrease in total cell nuclei area between 1h and 6h after *Mtb* infection for $PMN_{HIT}$ compared to $PMN_{HITTIN}$ ($p = 0.04$, pairwise t-test) (Figs 4 and S3).

When then comparing difference in NET area at 1h vs 6h, there was a statistically significant interaction between the PMN groups and infection status, $F(1, 10.5924) = 5.3398$, $p < 0.0421$). The simple main effect of phenotype group (considering the Bonferroni adjusted p-value) was significant for *Mtb* infection ($p = 0.0007$), but not for non-infection ($p = 1$). Consistent with the greater viability of $PMN_{HITTIN}$ at 6h of infection, pairwise comparisons show that $PMN_{HITTIN}$ also induce a significantly smaller change in NETs produced between 1h and 6 h of infection, compared to $PMN_{HIT}$ ($p = 0.0003$) (Figs 4 and S3). The transcriptional response for "Neutrophil extracellular trap formation" was driven by genes with significant positive FC (less downregulated), however the imaging data revealed overall lower NET formation in $PMN_{HITTIN}$. These findings were in line with the overall lower transcriptional response observed in $PMN_{HITTIN}$.

## DEGs in HITTIN vs HIT after 6h *Mtb* infection in the "Neutrophil extracellular trap formation" pathway

We next investigated the DEGs associated with the "Neutrophil extracellular trap formation" enriched in the combined KEGG pathway (Fig 3A, Table 3). In particular, the lower transcriptional and NET response in $PMN_{HITTIN}$ prompted further evaluation of genes which showed a lower upregulated response to *Mtb* in $PMN_{HITTIN}$ compared to $PMN_{HIT}$. Compared to $PMN_{HIT}$, *Mtb* infection of $PMN_{HITTIN}$, triggered a lower upregulation of genes involved in the multiple-protein NADPH oxidase complex including *Rac family small GTPase 2* (*RAC2*), and the transmembrane catalytic [cytochrome b-245 -alpha (*CYBA*) and -beta (*CYBB*)]. NADPH oxidase Nox2 (encoded by *CYBB*) and other cellular NADPH oxidases is involved with ROS production and NET formation [29–31]. *CYBB* is a nuclear factor kappa B (NF-κB) transcriptional target and together with *NFKB1 was also* less upregulated in $PMN_{HITTIN}$. Interestingly, *NCF2*, the gene encoding neutrophil cytosolic factor 2 (NCF-2 or p67-phox) was downregulated in $PMN_{HITTIN}$ and upregulated in $PMN_{HIT}$. $PMN_{HITTIN}$, also showed enrichment for DEGs related to "NADPH regeneration" (Fig 3A and 3B). NADPH can dually aid in ROS detoxification or production and is key for ROS mediated NET formation [30,32,33].

Histone deacetylases (HDACs) play a key role in NET formation and allow for peptidylarginine deiminase 4 (PAD4) mediated histone citrullination the initial step in chromatin decondensation [34,35]. Compared to $PMN_{HIT}$, $PMN_{HITTIN}$ had a less downregulated response in *HDAC1*, *HDAC3*, *HDAC4* and *PADI4* at 6h of *Mtb* infection. *Gasdermin D* (*GSDMD*) which plays a key role perforating the nuclear membrane to aid release of the decondensed chromatic during NET formation [36,37], also displayed the same pattern of expression regulation.

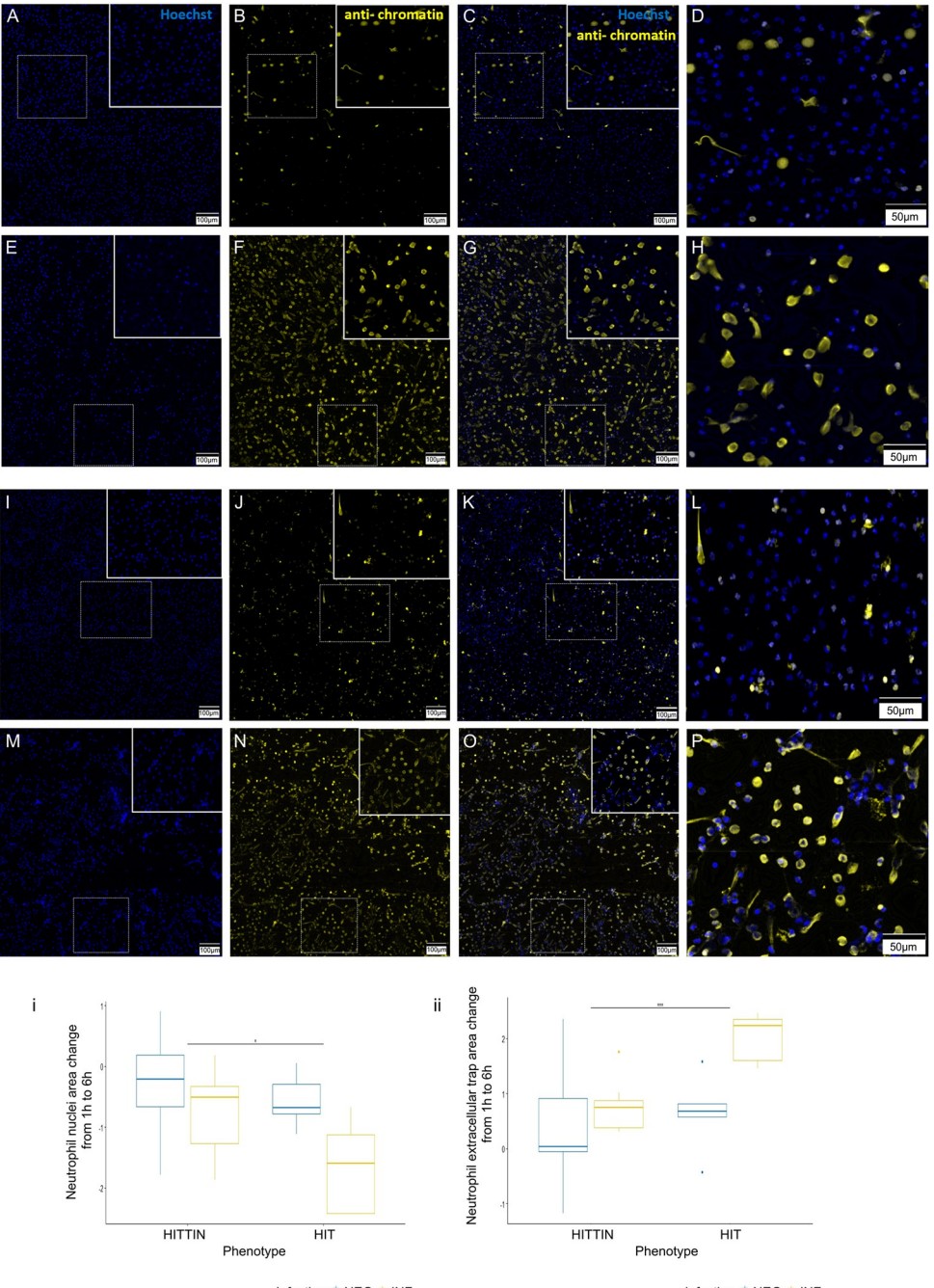

**Fig 4. Representative images and quantification of the change in NET and nuclei area following *Mtb* infection of neutrophils from HITTIN and HIT.** PMN$_{HITTIN}$ (A-H) and PMN$_{HIT}$ (I-P) following Mtb infection for 1h (A-D and I-L) and 6h (E-H and M-P). *Mtb* infected neutrophils at 1h and 6h were stained with Hoechst 33342 **(A, E, I, M)** and PL2-3 **(B, F, J, N)**. Overlap between the two stains is shown in (C, G, K, O) with enlarged box panels in **(D, H, L, P;** blue, DNA; yellow, chromatin); **(i)** Boxplots for total log change in nuclei area (Hoescht stain, calculated as shown in S1 Fig) from 1h to 6h uninfected and infected with *Mtb* in HITTIN vs HIT. There was a significant *Mtb* infection effect, F(1,12) = 9.729, p = 0.009 with pairwise comparisons showing a significant difference between the total change in cell nuclei area from 1h to 6h after *Mtb* infection in HITTIN compared to HIT (p = 0.04*, pairwise t-test), **(ii)** Boxplots for total log change in NET area (corrected for by the change in cell nuclei area, calculated as shown in S1 Fig) (PL2-3 anti-chromatin stain) from 1h to 6h uninfected and infected with *Mtb* in HITTIN vs HIT. There was a significant Mtb infection effect, F(1, 10.5924) = 5.3398, p < 0.0421 with pairwise comparisons showing a significantly difference between the total changed in cell nuclei area from 1h to 6h after *Mtb* infection in HITTIN compared to HIT (p = 0.0003***, pairwise t-test). The scale bars represent 100 μm **(A-C, E-G, I-K, M-O)** and 50 μm **(D, H, L, P)**.

*Caspase 1* (*CASP1*) and *4* (*CASP4*) which activate GSDMD have a lower upregulation in PMN$_{\text{HITTIN}}$ compared to PMN$_{\text{HIT}}$ after 6h *Mtb* infection [38].

Other DEGs enriched in the KEGG NET pathway, are also involved in additional neutrophil functional responses. Cell membrane receptors *TLR2* and *TLR4* were less upregulated in PMN$_{\text{HITTIN}}$ in response to 6h *Mtb* infection. Downstream of TLR4, pathway activation of NF-κB, Protein Kinase B (AKT) and phosphoinositide 3-kinase (PI3-K) lead to pro-survival mechanisms [39]. Integral to this TLR signaling system is mitogen-activated protein kinase (MAPK) and PI3-K. Dysregulation in especially the PI3-K/AKT signaling system contributes to an imbalance in neutrophil chemotaxis and can heighten inflammation and decrease pathogen clearance [40,41]. *MAP2K2* was less upregulated while *MAPK1*, *MAPK3*, *AKT1*, *AKT2* and *PIK3CD* were less downregulated in PMN$_{\text{HITTIN}}$. Azurocidin 1 (*AZU1*), the only antimicrobial peptide gene also included in the NET term, was downregulated after 6h *Mtb* infection in PMN$_{\text{HITTIN}}$ whilst upregulated in infected PMN$_{\text{HIT}}$ (Table 3).

## Discussion

HITTIN remain persistently TST, IGRA and TB negative despite prolonged *Mtb* exposure and antibody evidence of prior infection. This suggests HITTIN have different protective mechanisms in the early response to *Mtb* infection to that which occurs in HIT who convert to a positive TST and IGRA following *Mtb* infection. Here we investigated the DEGs of PMN$_{\text{HITTIN}}$ to determine whether they are a distinct and previously undefined group that contribute to HITTINs unique ability to control *Mtb* infection, prevent progression to TB, and interact with the adaptive immune response while possibly limiting persisting *Mtb*-specific interferon gamma (IFN-γ) T-cell memory responses. Using RNAseq analysis of *ex vivo Mtb* infected PMN we found that PMN$_{\text{HITTIN}}$ had an overall lower transcriptional FC response to *Mtb* after 6h of infection relative to PMN$_{\text{HIT.}}$ Positively enriched terms and pathways included apoptosis, NETosis, NADPH regeneration and ROS formation, with pathways related to necrotic cell death, necroptosis, neutrophil chemotaxis, degranulation and immune exhaustion which were triggered less strongly in PMN$_{\text{HITTIN}}$ compared to PMN$_{\text{HIT}}$ after 6h infection with *Mtb*. Using fluorescence microscopy, we demonstrate that after 6h of infection PMN$_{\text{HITTIN}}$ have undergone less NETosis than PMN$_{\text{HIT,}}$ corroborating the overall lower transcriptional response to *Mtb* infection after 6h seen in PMN$_{\text{HITTIN}}$. The question arises whether the less pronounced response by PMN$_{\text{HITTIN}}$ facilitates rapid *Mtb* control and a more contained neutrophil response, thereby preventing overt damage and an exacerbated inflammatory cascade, associated with TB. Their relatively lower induction of genes in response to *Mtb* suggests this is likely, requiring future functional studies for confirmation.

Neutrophil degranulation and NET formation processes were observed 1–6 months prior to persons developing TB and play an important role in *Mtb* infection progression to disease as well as potential lung destruction [42]. This highlights the important regulatory mechanism of NET formation by neutrophils in the inflammatory response against *Mtb* infection.

*Mtb*-induced ROS triggered necrosis in neutrophils and decreased the ability of macrophages to control *Mtb* growth [18]. Transcription of different groups of genes are regulated by NETosis-specific kinase cascades [43]. NETs are triggered by a NADPH/ROS dependent or an independent mechanism through TLR2/TLR4/lipopolysaccharide (LPS) activation [44,45]. The multi-protein NADPH oxidase complex consists of multiple subunits (including cytosolic p40-phox, p47phox and p67-phox, as well as the catalytic gp91-phox subunit) and other proteins (transmembrane p22-phox and nucleotide-binding Rac2) and is involved in the process to produce ROS [46]. ROS is mostly known to modulate a pro-inflammatory effect, with increased levels positively correlated to higher levels of TNF and a greater control of *Mtb*

infection [47]. High levels of ROS can be damaging, but at lower levels ROS can mediate an anti-inflammatory effect in mouse neutrophils by inhibiting p-AKT and NF-κβ and inducing apoptosis [48–53]. In our data, within the "Neutrophil extracellular trap formation" pathway in KEGG, several significantly under-stimulated genes in the pathway suggest potential relatively lower ROS levels in PMN$_{HITTIN}$ which could account for the relatively lower NET formation after 6h infection with *Mtb*. Another alternative mechanism PMN$_{HITTIN}$ displayed for inflammation control was enrichment of "NADPH regeneration". NADPH may play a role in neutralizing rather than producing ROS in PMN$_{HITTIN}$.

The lower upregulation of *CYBA*, *CYBB* (aka *NOX2*) as well as *RAC2* in PMN$_{HITTIN}$ could point to a mechanism to decrease ROS formation and prevent or decrease necrosis with consequently lower ROS mediated NET formation. Strikingly, *NCF2* was downregulated in PMN$_{HITTIN}$ and upregulated in PMN$_{HIT}$. The G allele of *NCF2* rs10911362 is associated with protection against TB in the Western Chinese Han population [54]. The rs3794624 polymorphism in *CYBA* was also linked with decreased TB susceptibility in Chinese Han persons [46,54]. PMN$_{HITTIN}$ exhibit an interesting potential *NCF2*-mediated mechanism driving less ROS and NET formation resulting in lower necrotic cell death.

TLR2/4 signalling could also mediate NET formation independent of ROS. A lower upregulation of *TLR2* and *TLR4* in PMN$_{HITTIN}$ likely balance potential inflammatory effects of NET release with downstream upregulation of apoptosis. NETs can also be triggered by activated CASP1 and ELANE cleaving GSDMD [55]. *CASP1* was less up- and *GSDMD* less downregulated in PMN$_{HITTIN}$. Although NETs have not been shown to control *Mtb* infection, the granular components associated with NETs could contribute [56]. Azurocidin 1 (AZU1) can interact with pentraxin 3 (PTX3) to enhance microbial function. The mechanism of this interaction is not known. PTX3 is a soluble pattern recognition receptor with increased levels seen in sepsis and TB [57]. *PTX3* was significantly upregulated by *Mtb* infection in both PMN$_{HITTIN}$ and PMN$_{HIT}$ but there was not a statistically significant difference between these. The fact that DEGs for other granule proteins were not identified is likely because they are mostly transcribed in immature neutrophils and this degranulation is reflective of their potentially different action [58–60].

HDAC1 has been implicated as a key regulator of innate immunity in monocytes isolated from HIV negative persons who tested persistently TST negative after household TB exposure [61]. HDAC inhibitors in macrophages such as phenylbutyrate improved *Mtb* control [61,62]. This inhibition was synergistically improved with vitamin D [62]. Less downregulation of *HDAC1* in PMN$_{HITTIN}$ translated into a decreased inflammatory response, with a lower upregulation of genes in the NF-kappa β signaling pathway (Fig 3A and 3C). This corroborates a more repressive or anti-inflammatory role in PMN$_{HITTIN}$.

Increased NETs can potentiate local damage due to release of granular components, but NET formation has also been shown to limit inflammation through degradation of chemokines and cytokines [63]. This intricate balance is likely maintained by PMN$_{HITTIN}$, which demonstrate lower NET formation in response to *Mtb* infection despite a positive enrichment of NET-related genes as compared to PMN$_{HIT}$. The formation of NETs itself is likely not problematic as PMN$_{HITTIN}$ could potentially effectively localize and trap *Mtb* through NETs and activate other innate cells such as macrophages [22]. The balance of NETosis with other cell death mechanisms may play an additional role to determine the fate of *Mtb*.

## Study limitations

Use of ART can significantly affect the transcriptional responsiveness of alveolar macrophages to *Mtb*. It is important to note that although HITTIN were on ART for a shorter time, there

was no significant difference to the time spent on ART in HIT and therefore ART is an unlikely factor driving the transcriptional differences observed between neutrophils from HIT-TIN and HIT [64].

The effect of contaminating T cells in PMN cultures cannot be completely excluded. T cells could possibly contribute to IFN-γ driven PMN transcriptional differences with IFN-γ known to upregulate key genes affecting NADPH activity [65,66]. Per clinical classification, HITTIN lack IFN-γ T cell responses as measured by IGRA. In addition, good PMN purity (90%) vs 5% in T cells, as well as the 6h timepoint which is short for the IFN-γ driven responses, argue against the significant impact of contaminating cells. Further classification of non IFN-γ T cell subsets in HITTIN is needed.

To account for large inter-individual differences in response to *Mtb*, we employed subject-specific fold changes. This approach precludes comparisons of group expression levels. Future work is needed to determine if there is a measured difference in ROS released by $PMN_{HITTIN}$ compared to $PMN_{HIT}$. In addition, neutrophil *Mtb* killing assays will determine if $PMN_{HITTIN}$ show improved *Mtb* infection clearance. Multiple studies have identified neutrophil subpopulations and it is possible that a specific subpopulation could be driving the response differences we observed [67–69]. Flow cytometry for identification of potential subpopulations as well as single cell RNA sequencing would be highly informative.

## Conclusion

In general, for TB, neutrophils are mostly linked to hyperinflammatory responses and more severe disease. Here we showed a distinctive gene expression profile for neutrophil responses from HITTIN individuals, who appeared protected from TB despite a lack of canonical *Mtb*-specific T-cell memory. These findings put neutrophils at the forefront of potential innate immune cell mechanisms of *Mtb* infection resistance. They highlight a distinct phenotypic response to *Mtb* in $PMN_{HITTIN}$ compared to neutrophils from persons otherwise defined as sensitized by *Mtb*.

We measured a reflection of the overall impaired transcriptional response and consequently no claim can be made about specific effect. Rather, we are reporting a less pronounced response to *Mtb* infection in HITTIN after 6h that also effects pathways known to represent anti-mycobacterial activity. Contrary to *Mtb* inducing necrotic cell death mechanisms as in most neutrophils, DEGs in $PMN_{HITTIN}$ showed decreased transcriptional responses for necrosis, with an enrichment of terms related to apoptotic cell death and NETosis. Fluorescence imaging corroborated significant reduction in NET formation between $PMN_{HITTIN}$ and $PMN_{HIT,}$ likely driven by lower ROS transcriptional pathways with a downregulation of *NCF2*, a key mediator in ROS formation, in $PMN_{HITTIN}$. These molecular data implicated neutrophils as key effector cells in *Mtb* infection resistance. Further increased understanding of the crucial pathways of *Mtb* infection control highlighted in the study could be harnessed for the development of *Mtb* prevention and treatment strategies.

## Materials and methods

### Ethics statement

The study was approved by the Health Research Ethics Committee of Stellenbosch University (S18/08/175(PhD)). Samples used in this study were leveraged from the ResisTB study. The Health Research Ethics Committee of Stellenbosch University (N16/03/033 and N16/03/033A) and the Faculty of Health Sciences Human Research Ethics Committee of the University of Cape Town (755/2016 and 702/2017) approved participant recruitment for ResisTB. Additional approval was obtained from the City of Cape Town and Western Cape government for

access to the relevant clinics. Formal consent for research participation was given by each participant by signing written informed consent documents.

## Participant recruitment

Samples were collected from 29 participants recruited to the ResisTB study in Cape Town, South Africa. The ResisTB study recruited a group of participants between the ages of 35–60 years old. The recruitment of this group for the ResisTB study has been fully described previously [4]. Briefly, the participants had to be living with HIV in an area of high *Mtb* transmission and have no history of previous or current TB. They had to have a history of living with a low CD4+ count (either with two CD4+ <350 cells/mm$^3$ counts at least 6 months apart or a single CD4+ count <200 cells/mm$^3$) prior to initiating antiretroviral therapy (ART), and be immune reconstituted on ART for at least one year at time of enrolment with the last CD4 + count >200 cells/mm$^3$ [4]. Exclusion criteria included pregnancy, previous TB, symptoms suggestive of active TB disease, participation in other interventional studies, and any AIDS defining illness in the year prior to enrolment. Participants were seen at three visits. During the enrolment visit, blood samples were taken for an IGRA using the QuantiFERON-TB Gold Plus (QFT-Plus) in tube test. For follow up, whole blood was collected in an EDTA tube for Ficoll gradient separation and neutrophil isolation as well as a second IGRA, followed by TST administration with PPD RT23 (Staten Serum Institute). After 3 days the TST reading was taken and participants who tested IGRA negative from the first visit had bloods taken for a third and final IGRA. This ensured that any T-cell response that would be boosted by TST administration would be identified by IGRA to ensure participants with low level T-cell memory were correctly identified.

For this study, samples were leveraged from 17 older HITTIN and 11 HIT participants. Age was used as a surrogate for increased exposure frequency to *Mtb* since most persons are infected with *Mtb* by the age of 30–35 in the Western Cape of South Africa, where the study was conducted [70,71].

## Neutrophil isolation

Neutrophils were isolated by Ficoll gradient separation. Whole blood diluted 1:1 with 1x Phosphate buffered saline (PBS, Sigma-Aldrich, USA) was layered over the density gradient separation medium (Histopaque / Ficoll-Paque, Sigma-Aldrich, USA). Cells were centrifuged for 25 minutes at 400 x g. After this, cells were washed twice with 4˚C PBS and centrifuged each time at 400 x g for 10 minutes at 4˚C. Peripheral blood mononuclear cells were removed first and then the remaining plasma and Ficoll-Paque layer. A red blood cell (RBC) lysis buffer (component concentrations) was added in a ratio of 1:10 or topped up to 50ml, if a 50 ml centrifuge tube was used, to the remaining bottom layer and incubated for 10 minutes at 4˚C. RBC lysis buffer (8% Ammonium chloride [NH$_4$Cl], 0.8% Sodium bicarbonate [NaHCO$_3$, Sigma-Aldrich, USA] and 0.4% Ethylenediaminetetraacetic acid [EDTA, Sigma-Aldrich, USA]).

After RBC lysis, samples were centrifuged at 400 x g for 10 minutes at 4˚C and then washed twice with PBS (4˚C) and centrifuged at 400 x g for 10 minutes at 4˚C. After the cell count, 1 x 10$^7$ cells in RPMI-1640 with L-glutamine and sodium bicarbonate (Sigma-Aldrich, USA), were seeded evenly over each row of 3 wells in a 6-well plate cell culture plate (Nest Scientific USA Inc., USA) at 0.33 x 10$^7$ cells per well for the RNA sequencing experiment and 2 x 10$^5$ cells per well in a 96-well (M0562, Greiner, Sigma Aldrich, USA) for microscopy. For each participant two 6-well plates (one for each time-point i.e., 1h and 6h, with 3 wells for no infection and 3 wells for *Mtb* infection) were seeded. Cells were incubated for 1h-2h at 37˚C and 5% CO$_2$. An aliquot of remaining neutrophils was fixed with 4% Paraformaldehyde (PFA, cat.no.

43368, Alfa Aesar, USA) and then stored overnight at -80˚C before transferred to and stored in liquid nitrogen.

## Staining for flow cytometry

The fixed neutrophil aliquots were thawed and washed in Dulbecco's phosphate buffered saline (DPBS, Cat. No. 14190144, Thermo Fisher Scientific, Australia). 1.5 x $10^6$ cells per participant were resuspended in surface staining buffer comprised of DPBS + 3% Foetal Bovine Serum (FBS, Cat. No. SFBS-AU, Bovogen Biologicals Pty Ltd, Australia). Fluorescently conjugated antibodies for cell surface staining (S1 Table) were prepared in Brilliant Stain Buffer (Cat. no. 556349, BD Biosciences, Australia) and incubated with cells in a 96 well plate (Cat. no. COR3894, Corning, In Vitro Technologies Pty Ltd, Australia) for 30 minutes at room temperature. Cells were washed using surface staining buffer, permeabilised using a 10-minute incubation in Perm/Wash Buffer (P/W, cat. no. 554723, BD Biosciences, Australia) and washed once with P/W. The remaining antibodies listed in S1 Table were prepared in Brilliant Stain Buffer and incubated with cells in a 96 well plate for 45 minutes at 4˚C. Cells were washed twice with P/W and flow cytometry data was acquired using the Cytek Aurora Spectral Flow Cytometer (5 laser, 64 detector configuration). Data was unmixed using SpectroFlo (Version 3.0.3) and analysed using FlowJo (FlowJo 10.8.2) and GraphPad Prism 8.0.1 (see methods section *Flow data analysis*, S4 Fig). Cell single-stained controls were prepared using the same protocol as experimental samples.

## Mycobacterial cultures and neutrophil *in vitro* infection

*Mtb* single cells stocks for infection were prepared as previously described [62]. *Mtb* H37Rv was grown in a liquid culture of Middlebrook 7H9 medium (Difco, Becton Dickinson, USA) with albumin-dextrose-catalase (ADC, Becton Dickinson, USA) and 0.05% Tween-80 (Sigma-Aldrich, USA) at 37˚C as a standing culture in a tray and mixed by swirling every few days to disperse clumps for 10 days. After 10 days, a liquid culture of 7H9 ADC, without Tween-80, was inoculated with 1/100th volume of day 10 end-exponential growing phase *Mtb* and incubated in static standing culture, only swirled periodically, at 37˚C for 10 days. After this, cultures were centrifuged for 5 minutes at 2500RPM. Glass beads were used to break the pellet after which it was resuspended in PBS. The upper part of the bacterial suspension was harvested and spun for 10 minutes at 1400RPM. Then the upper part of the bacterial suspension was harvested again and mixed with glycerol (5% final volume) and aliquots were stored at -80˚C. Aliquots was serial diluted before and after freezing and plated on Middlebrook 7H10 agar (Becton Dickinson, USA) plates with oleic acid-albumin-dextrose-catalase (OADC, Becton Dickinson, USA) for colony forming unit (CFU) determination. Prior to infection, aliquots were thawed at room temperature.

Neutrophils were infected at a multiplicity of infection (MOI) of 1:1 for 1h, and 6h at 37˚C under 5% $CO_2$. For the 3 wells in row 1 of each of the two plates, 1ml was removed from each well (3ml per plate giving a total of 6ml from the two plates). After this 2 x $10^7$ *Mtb* H37Rv was added to the 6ml, carefully pipetted to mix and then 1ml (0.33 x $10^7$ *Mtb* H37Rv) was returned to each of these wells for infection. For each infection experiment, *Mtb* infection MOI were confirmed by CFU counts of serially diluted inoculum plated on 7H10 OADC agar.

## Staining for microscopy

Of the neutrophils isolated from 17 older HITTIN and 11 HIT participants, neutrophils from 12 HITTIN and 10 HIT were plated and stained for microscopy imaging. Cell staining was performed on 2 x $10^5$ neutrophils per well for uninfected vs infected with *Mtb* H37Rv for each

timepoint 1 and 6h. For each time point (1h, 6h) two wells (A, B) were plated for neutrophils to be infected with *Mtb* and two wells for no infection (C,D). At the designated timepoints the supernatant was discarded and replaced with 4% PFA (cat.no. 43368, Alfa Aesar, USA). The plates were incubated at 4˚C for 24h, before removing the plates from the BSL-3. PFA was replaced with PBS and plates stored at 4˚C.

For immunofluorescence labelling, wells were stained with 1˚ [1:500 mouse mAB PL2/3, kindly gifted by Arturo Zychlinsky (nucleosomal complex of Histone 2A, Histone 2B and chromatin)] and 2˚ antibody cocktail [1:1000 α-mouse-Cy3 (cat.no. 715-166-150, Amersham, UK) and 10µg/ml Hoechst 33342 (cat.no. 14533, Sigma-Aldrich, USA)] [72–74]. Prior to staining PBS was removed from all wells and replaced with Perm/Quench Buffer (50mM NH4Cl, 0.2% saponin in PBS) for a 15-minute incubation period. After removing Perm/Quench, PGAS Buffer (0.2% bovine serum albumin (BSA), 0.02% saponin and 0.02% Azide (NaN$_3$) was added for a 5-minute incubation. Once the PGAS was removed the 1˚ antibody cocktail was added and left to incubate overnight at 4˚C. The following day the 1˚ antibody cocktail was discarded, and two washes were completed with PGAS buffer before adding the 2˚ antibody cocktail and incubating in the dark for 1h at room temperature. The staining was completed by removing the 2˚ antibody cocktail and completing three washes with PGAS buffer. Each well was filled with PGAS buffer to the brim. The plates were covered with foil and stored at 4˚C for imaging.

## Image acquisition

Image tile scans were acquired with the Zeiss AxioObserver Z1 microscope, equipped with a Colibri 7 light source for excitation of Hoechst 33342 (cat.no. 14533, Sigma-Alrich) with LED-module 385nm, and Cy3 with LED-module 511nm. A quadruple band pass filter and triple band pass filter were used respectively for detection of Hoechst (wavelength range 412-438nm) and Cy3 (wavelength range 546–564 nm). Images were acquired with a LD A-Plan 40x/0.55 objective as a 6 x 8 tile scan to acquire a total area of 1.21mmx1.21mm (S1 Fig).

## Imaging processing

Tiles were stitched together into single images using Zeiss ZenPro software (version 2.6), which were imported into FIJI/ImageJ (version 1.53t). Image tiles were split into separate channels, after which background subtraction (ranging from 50–100 pixels) and Otsu based thresholding was conducted on a per image basis to generate a binarized map of each image. Binary images were further processed by using the binary closing function in FIJI, as well as a top hat filter of 1–2 pixels to remove small non-specific pixels. For the nuclear quantification, an additional watershed function was applied to separate borders more accurately. Thereafter, morphometric data were obtained by specifying the area and circularity ranges of particles to be analysed through the Analyze Particles function in FIJI to determine the total area covered.

## Preparation of RNASeq libraries

Uninfected and infected neutrophils (1x10$^7$ cells for each) were lysed with TRIzol (Invitrogen TRIzol Reagent, Fisher Scientific, USA) after 1h and 6h after infection, and stored at -80˚C. The miRNeasy kit (Qiagen, Germany) was used for total RNA extraction. One sample per participant and condition was used. RNA integrity (RIN) was assessed with the Agilent 2100 Bioanalyzer (Agilent Technologies, Germany). Samples with RIN >7 were selected for library preparation using TruSeq RNA Library Preparation Kit v2, Set A (Illumina, USA). Samples were sequenced in two batches (S4 Table). Batch 1 was a preliminary test batch for exploratory analysis and to confirm data quality from PMN, consisting of samples from 3 HITTIN (18%)

and 4 HIT (36%) participants. This was sequenced as unstranded, 100bp, single-end (SE) on an Illumina HiSeq4000 sequencer at Genome Quebec, Montreal, Canada. Batch 2 with samples from 14 HITTIN (82%) and 7 HIT (64%) participants was sequenced as unstranded, 150bp paired-end (PE) on an Illumina NovaSeq6000 sequencer at Genome Quebec, Montreal, Canada. There was no significant difference in the distribution of the samples used for each batch (p = 0.4, Fisher's exact test) (S4 Table).

### Quality control and raw data pre-processing

The quality of the raw sequence data was accessed by FastQC (v0.11.5) and MultiQC [75,76]. The mean quality of reads was high with the mean sequence quality score (Phred Score) >35 for both batches. Duplicates were observed in both batches. For batch 1, sequenced as single-end reads (SE), we detected a fraction of 0.41–0.63 duplicates which fall within and below the expected range (0.66–0.74) of duplicates according to the Universal Human Reference RNA (UHRR) [77]. The fraction of duplicates (0.4–0.61) from batch 2 was much higher than expected (0.087–0.18) for paired-end (PE) reads [77]. With single-end reads, fragmentation bias is usually the cause of these duplicates. For both SE and PE, de-duplication is not recommended [77]. To minimize bias, duplicates were not removed from either of the batches.

Batch 1 and 2 were combined after read counts were generated. For read count generation the same method was applied to both batches. After initial raw sequence quality check by FastQC, data was filtered and trimmed using HTStream [75,78]. The occurrences of rRNA read contamination was counted but not removed and contamination with PhiX, a control in Illumina runs, was removed by hts_SeqScreener [78]. Adapters were trimmed with hts_AdapterTrimmer and poly(A) tails were removed with hts_PolyATTrim [78]. Any remaining N characters (unassigned bases) were removed with hts_NTrimmer and hts_QWindowTrim for quality trimming the end of the reads [78]. Reads less than seventy-five base pairs long were removed by hts_LengthFilter [78].

The genome was indexed using GRCh38.p13 v34 (ENSEMBLE v100) and reads were aligned to the genome with STAR (v2.5.3a) in a bam format [79–81]. More than 80% of the reads were uniquely mapped. The reads per gene output files were combined before the final unstranded read counts matrix was extracted for all. The gene read count matrix was input in R (v4.0.3) for further analysis [82]. The untransformed and raw count matrix was adjusted for batch effect using ComBat-seq while preserving the signal of the biological variables of interest namely the phenogroups (HITTIN vs HIT), the timepoints and infection [83] (S5 Fig).

### Differential gene expression analysis

Raw counts were transformed to counts per million (CPM) and filtered using "filterByExpr" in the edgeR package [84]. Briefly, the function kept genes at least 25 read counts or more in at least a minimum of samples (calculated as 70% of the samples in the smallest group). For this dataset genes with a CPM of 1.32 in at least 8 (70% of the smallest group of 11 samples) are retained, leaving 14602/60622 (24%) of the genes for further differential expression analysis (S6 Fig). Normalization scaling factors were generated by "calcNormfactors" in edgeR using the method of trimmed mean of M-values (TMM) [84].

Outlier samples were observed in the multidimensional scaling (MDS) plots (S5 Fig). Instead of removing the outliers, limma's function "voomWithQualityWeights" allows for variations in samples by taking sample-specific variablity as well as the 'global intensity-dependent variability trends' as accounted for by 'voom', into consideration [85]. The expression matrix was normalized and transformed to $\log_2$ CPM and sample specific weights were

incorporated with abundance dependent weights using limma's (v3.46.0) function "voom-WithQualityWeights" [85,86] (S7 Fig).

## Data exploration

Data was visualized with multidimensional scaling (MDS) plots and principal components analysis (PCA) Scree plot (S8 and S9 Figs). Normalized counts were plotted without covariate correction to examine data for potential covariate effect. Initially clear separation was seen by batch effect, but this was corrected for (S5 Fig). The post batch corrected data was reviewed (S10–S17 Figs). Sex and smoking showed some separation and would need further investigation (S10 and S11 Figs). No separation could be seen for age, BMI, chronic disease history, previous INH used, duration of INH and alcohol use (S12–S17 Figs). Using the paired design with blocking, accounted for the covariates effects by blocking on the effect of each individual and are discussed below.

## Model design

For the analysis we used a paired design and created the model with subject IDs (for blocking in a paired design), group (HITTIN and HIT), time of infection (1 and 6h) and *Mtb* infection status (uninfected and infected) as factors in the model using model.matrix from the stats package (v4.0.3).

In the individual blocking, individuals were defined per timepoint, due to large variance introduced to each individual due to time effect (S8 Fig). The three factors group, time of infection and *Mtb* infection status were grouped together as a single interaction term and was modelled with the subject effect as a means model:

$$E(Gi)_{Model1} \sim \beta_0 + \sum_{j=1}^{n} \beta_{jkl} x_j + \beta_{Hkl}.x_{Hkl} + \beta_{Nkl}.x_{Nkl} + \epsilon$$

Where Gi is the $\log_2$(CPM) expression for each gene i (n = 14602) and E(Gi) is the expected gene expression. $\beta_0$ represents the intercept. $\beta_{jkl}$ represents the mean expression for each individual j (j = 1 to total of n) at time k (k = 1h or 6h) for *Mtb* infection state l (l = uninfected or infected). H represents HIT and N represents HITTIN. $x_j$ is a variable representing the n samples in the data. The $\beta$ value is the mean expression for each specified group after blocking on the individual averages. $\epsilon$ is the residual term and is assumed to be normally distributed with a constant variance across the range of data.

Contrasts were made using makeContrasts and defined as:

i) Group specific *Mtb* infection compared to no infection effect at 1h
$(\beta_{HITTIN1inf}, \beta_{HIT1inf})$

ii) Group specific *Mtb* infection compared to no infection effect at
6h $(\beta_{HITTIN6inf}, \beta_{HIT6inf})$

iii) Differential response between groups at 1h infected

$(\beta_{HITTIN1inf}) - (\beta_{HIT1inf})$

iv) Differential response between groups at 6h infected

$(\beta_{HITTIN6inf}) - (\beta_{HIT6inf})$

Linear models were fitted with lmFit and eBayes to calculate the gene-wise test statistics (moderated t-statistic, p-values and B-statistic). Criteria for DEGs was defined by a Benjamini Hochberg FDR procedure as genes with an absolute logFC $\geq$ 0.2 and adjusted p value $\leq$ 0.05.

DEGs were used in a gene set enrichment analysis for pathways and Gene ontology (GO) enrichment using ReactomePA v1.34.0 and clusterProfiler v3.18.1 [87]. enrichGO was used to

test GO biological process, enrichKEGG for KEGG pathways and Reactome with enrichPathway. An FDR $\leq 0.1$ was used for Benjamini- Hochberg's multiple testing correction. Pathways and GO terms with less than five assigned genes were excluded.

### Flow data analysis

CD15+CD66b+ and CD15-CD66b- cells were calculated as a percentage of total CD45+ single cells. Each subset CD15+CD66b+ and CD15-CD66b- was then further stratified into the relevant subpopulation contribution, which were then expressed as a percentage of each subset. CD15+CD66b+ was stratified as CD16+ (neutrophils) and CD16-CD14$_{low}$ (eosinophils), and CD15-CD66b- into CD3+ (T-cells), CD3- CD14+ (Monocytes) and CD3-CD14- (Other). Wilcoxon rank sum test was used to calculated if there was a significant ($p<0.05$) difference in the median percentage contribution of each subset in HITTIN vs HIT using GraphPad Prism 8.0.1 (S2 Table).

### Fluorescence image analysis

Image from total of 9 HITTIN and 5 HIT neutrophil were included in the final analysis. Results of the Hoechst cell nuclei area count as well as the total NETs area (chromatin channel) were analysed in R studio. The log ratio of total nuclei area (Hoechst channel) at 6h to 1h was used to compare the infection effect of *Mtb* between and within groups on the change in area of cells (as a proxy for cell counts, since masking often under or overestimated cell counts, especially with neutrophil nuclei with trinucleate structures and cell clumps) over 1h to 6h (Figs 4i and S1). The total NET area change was determined by the total NET area (chromatin channel) to the total cell area (Hoechst channel) as determined for each timepoint and non-infection or infection with *Mtb*. To investigate if there is a difference in NET area change from 1h to 6h between PMN$_{HITTIN}$ and PMN$_{HIT}$ in response to infection, we calculated the log transformed ratio of the NET area change at 6h to 1h in uninfected and *Mtb* infected for each subject (Figs 4ii and S1).

A two-way mixed ANOVA analysis approach was used with the phenotype group (HITTIN vs HIT) describing the between-subject factor and *Mtb* infection status (non-infected vs infected) the within subject factor. Assumptions of normality, homogeneity of variances and homogeneity of covariances were met. After the removal of the initial extreme outlier, there were still some 3 outliers in the remaining analysis. These outliers were not removed, and a robust ANOVA was performed using the WRS2 package in R [88]. Further pairwise comparisons were done between PMN$_{HITTIN}$ and PMN$_{HIT}$ for non-infection as well as infection with *Mtb*, using a pairwise t-test and Bonferroni adjusted p-values.

## Supporting information

**S1 Table. Antibodies used for Flow Cytometry Analysis of Contaminating Cell Populations.**
(PDF)

**S2 Table. Cell population distribution of isolated PMN from HITTIN and HIT, as determined by flow cytometry.**
(PDF)

**S3 Table. Differential gene expression testing results.**
(PDF)

**S4 Table. Sample characteristics.**
(PDF)

**S1 File. Table showing the GO, KEGG and Reactome results for the interaction as well as individual group *Mtb* infection responses after 6h.**
(XLSX)

**S2 File. Table showing cell population distribution data as determined by flow cytometry.**
(XLS)

**S3 File. Table showing morphometric data from microscopy.**
(XLS)

**S1 Fig. Volcano plots of differential gene expression at 1h infection by PMN from HITTIN and HIT.** Volcano plot for transcriptional responses to *Mtb* challenge for neutrophils from HITTIN ($PMN_{HITTIN}$) and HIT ($PMN_{HIT}$) participants at 1h **(A-C)** post *Mtb* infection. The y-axis shows the negative log10 unadjusted P value and the x-axis the $\log_2$ fold change (FC). The vertical dashed lines represent $\log_2$ FC thresholds of -0.2 and 0.2. Each gene is represented by a dot. Genes which are downregulated or upregulated as determined by the FDR $\leq$ 5% are shown in blue and red, respectively. Genes with non-significant expression changes and below the $\log_2$ FC threshold are shown in grey. Differentially expressed genes at 1h post-infection compared to 1h uninfected PMN from HITTIN ($PMN_{HITTIN}$) **(A)** and HIT participants ($PMN_{HIT}$) **(B)**. Significant differentially triggered genes between $PMN_{HITTIN}$ and $PMN_{HIT}$ at 1h post infection **(C)**.
(PDF)

**S2 Fig. Correlation plots of $\log_2$FC gene expression after *Mtb* infection of PMN from HITTIN and HIT.** "HITTIN 1h infected" represents the $\log_2$FC gene expression effect when comparing the 1h *Mtb* infection to the 1h uninfected response in neutrophils (PMN) from HIV-1-positive persistently TB, tuberculin and IGRA negative individuals ($PMN_{HITTIN}$)(x-axis). "HITTIN 6h infected" is the same but for 6h. "HIT 1h infected" represents the $\log_2$FC gene expression effect when comparing the 1h Mtb infection to the 1h uninfected response in PMN from HIV-1-positive, IGRA positive, tuberculin positive individuals ($PMN_{HIT}$)(y-axis). "HIT 6h infected" is the same but for 6h. Scatterplots display the correlation of the $\log_2$FC gene expression as defined for each group at 1h **(A and B)** or 6h **(C and D)**. Each grey dot represents a differentially expressed gene (DEGs). The grey dots show combined DEGs from both groups represented on the x and y axis. Blue coloured dots indicate DEGs from the x-axis group and red dots indicate the y-axis group. Pearson correlation (R value) was calculated for the coloured DEGs. The $\log_2$FC for each group is plotted with the y-axis as the reference group. Differential gene expression is more pronounced in genes with the highest $\log_2$FC, with the majority of genes displaying good correlation between phenotype groups.
(PDF)

**S3 Fig. Fluorescence imaging process overview to calculate neutrophil nuclei and neutrophil extracellular trap (chromatin) area. Scale bars: 100 μm (A-C) and 50 μm (D-I).** Isolated neutrophils were stimulated with *Mtb* and stained with Hoechst 33342 **(A and boxed area D)** and PL2-3 **(B and boxed area E)**. **C and the boxed area F** show the overlap (blue, DNA; yellow, chromatin). The segmentation of fluorescent signals used to calculate the neutrophil nuclei area **(G)** and neutrophil extracellular trap area **(H)** with the overlap shown in **(I)**. The scale bars represent 100 μm **(A-C)** and 50 μm **(D-I)**.
(PDF)

**S4 Fig. Flow Cytometry Analysis of Cell Populations.** The first gate was applied to exclude debris (the low SSC-H, FSC-H values in the bottom left corner). Single cells were separated and gated by a CD45+ marker for leukocytes. After single cell gating, CD45+ cells were then grouped into CD15+CD66b+ (granulocytes) and CD15-CD66b-(non-granulocytes) cells. The CD15+CD66b+ cells were further classified as CD14- CD16+ (Neutrophils) and CD14- CD16- (Eosinophils). CD15-CD66b- were stratified as CD3+ (T-cells), CD3- CD14+ (Monocytes) and CD14- CD3- (Other) cells.
(PDF)

**S5 Fig. Multidimensional scaling (MDS) plot before (A) and after (B) ComBat-seq batch correction.** The multidimensional scaling (MDS) plots the Euclidian distances between samples with the x and y axis representing the sample distances between samples of read counts normalized by depth but not covariates. Each row of plots in **(A)** and **(B)** represent dimension 1 to 5 respectively (represented by the x-axis) and shown with the combination of the other dimensions on the y-axis. Samples are colored for batch effect with green showing samples sequenced on Illumina HiSeq2500 and blue for samples sequenced on Illumina NovaSeq6000. **(A)** shows the multiple combination of plots by dimension for the normalized counts prior to batch correction. Clear separation of samples by batch can be seen and is most notable in dimensions 4 and 5. **(B)** Appropriate batch correction after correction of raw counts with ComBat-seq. After batch correct outlier samples are most notably observed in dimension 3.
(PDF)

**S6 Fig. Density plot showing successful filtering applied to counts.** Density plot with the density represented on the y-axis for the log-cpm values before **(A)** and after **(B)** filtering. The dotted vertical line is equivalent to the counts per million (CPM) threshold of 1.32 which was used in the filtering step.
(PDF)

**S7 Fig. Voom observational mean-variance trend and sample-specific weights. (A)** shows rescaled (square-root of standard deviations) residual variances plotted against the mean expression ($\log_2$ transformed with an offset of 2) of each gene. A decreasing trend is seen between the mean gene expression and the variance with higher expressed genes showing less variation. The sample specific weight for each sample used in the analysis is shown in **(B)**. A total of 112 samples were analysed (28 participants and 4 conditions, uninfected and infected after 1 and 6h).
(PDF)

**S8 Fig. Multidimensional scaling (MDS) plot of group separation.** The multidimensional scaling (MDS) plots the Euclidian distances between samples with the x and y axis representing the sample distances between samples of read counts normalized by depth but not covariates. Each row of plots in represent dimension 1 to 5 respectively (represented by the x-axis) and shown with the combination of the other dimensions on the y-axis. Samples are colored for phenotype (HITTIN and HIT), timepoint (1h and 6h) and infection status (uninfected [NEG] and infected [INF]) and as depicted in the legend. Separation of the groups by time can be seen in dimension 1, by infection in dimension 2 and the phenotype in dimension 3. Additional separation is seen in dimension 4 and 5 due to sex and possibly smoking (see S8 and S9 Figs).
(PDF)

**S9 Fig. Scree plot.** The top 5 principal components are represented on the x-axis and the variance explained by each component on the y-axis. Principal component (PC) 1 contributes to

most of the variance seen and represents time. PC2 represent infection effect and PC3 the phenotype groups.
(PDF)

**S10 Fig. Multidimensional scaling (MDS) plot of participant sex.** The multidimensional scaling (MDS) plots the Euclidian distances between samples with the x and y axis representing the sample distances between samples of read counts normalized by depth but not covariates. Each row of plots in represent dimension 1 to 5 respectively (represented by the x-axis) and shown with the combination of the other dimensions on the y-axis. Samples are colored for sex (female and male) as depicted in the legend. Separation is seen in dimension 4 and 5.
(PDF)

**S11 Fig. Multidimensional scaling (MDS) plot of participant smoking habit.** The multidimensional scaling (MDS) plots the Euclidian distances between samples with the x and y axis representing the sample distances between samples of read counts normalized by depth but not covariates. Each row of plots in represent dimension 1 to 5 respectively (represented by the x-axis) and shown with the combination of the other dimensions on the y-axis. Samples are colored for participants classified as smokers or not as depicted in the legend. Each participant is represented by 4 dots (1h uninfected and infected; 6h uninfected and infected). There are only two smokers making it difficult to comment on separation due to smoking.
(PDF)

**S12 Fig. Multidimensional scaling (MDS) plot of participant isoniazid (INH) prophylaxis use.** The multidimensional scaling (MDS) plots the Euclidian distances between samples with the x and y axis representing the sample distances between samples of read counts normalized by depth but not covariates. Each row of plots in represent dimension 1 to 5 respectively (represented by the x-axis) and shown with the combination of the other dimensions on the y-axis. Samples are colored for participants classified as either having used INH prophylaxis previously/currently or not, as depicted in the legend. There is no clear separation based on INH prophylaxis use.
(PDF)

**S13 Fig. Multidimensional scaling (MDS) plot of duration of participant isoniazid (INH) prophylaxis use in months.** The multidimensional scaling (MDS) plots the Euclidian distances between samples with the x and y axis representing the sample distances between samples of read counts normalized by depth but not covariates. Each row of plots in represent dimension 1 to 5 respectively (represented by the x-axis) and shown with the combination of the other dimensions on the y-axis. Samples are colored for the numbers of months participants previously used INH (0 or none, 2, 6, 12, 24 or 36 months) or participants who are currently using INH, as depicted in the legend. There is no clear separation based on duration of INH prophylaxis use.
(PDF)

**S14 Fig. Multidimensional scaling (MDS) plot of participants with a known history of chronic disease.** The multidimensional scaling (MDS) plots the Euclidian distances between samples with the x and y axis representing the sample distances between samples of read counts normalized by depth but not covariates. Each row of plots in represent dimension 1 to 5 respectively (represented by the x-axis) and shown with the combination of the other dimensions on the y-axis. Samples are colored for participants classified as either having a chronic disease or not, as depicted in the legend. There is no clear separation based on participants

with chronic disease.
(PDF)

**S15 Fig. Multidimensional scaling (MDS) plot of participant social alcohol use.** The multi-dimensional scaling (MDS) plots the Euclidian distances between samples with the x and y axis representing the sample distances between samples of read counts normalized by depth but not covariates. Each row of plots in represent dimension 1 to 5 respectively (represented by the x-axis) and shown with the combination of the other dimensions on the y-axis. Samples are colored for participants classified as either using alcohol or not, as depicted in the legend. There is no clear separation based on participant alcohol use.
(PDF)

**S16 Fig. Multidimensional scaling (MDS) plot of participant body mass index (BMI).** The multidimensional scaling (MDS) plots the Euclidian distances between samples with the x and y axis representing the sample distances between samples of read counts normalized by depth but not covariates. Each row of plots in represent dimension 1 to 5 respectively (represented by the x-axis) and shown with the combination of the other dimensions on the y-axis. Samples are colored for participants classified according to their BMI, as depicted in the legend. NA represents samples with missing values and is shown in grey. There is no clear separation based on participant BMI.
(PDF)

**S17 Fig. Multidimensional scaling (MDS) plot of participant age.** The multidimensional scaling (MDS) plots the Euclidian distances between samples with the x and y axis representing the sample distances between samples of read counts normalized by depth but not covariates. Each row of plots in represent dimension 1 to 5 respectively (represented by the x-axis) and shown with the combination of the other dimensions on the y-axis. Samples are colored for participants classified by age, as depicted in the legend. There is no clear separation based on participant age, although there is an older outlier seen in dimensions 3.
(PDF)

## Acknowledgments

Special acknowledgement to Dr Tracey Jooste for her manuscript input as well as Dr Sihaam Boolay and Dr Nasiema Allie for laboratory assistance and support and Arturo Zychlinsky for the kind supply of PL2/3 mAb.

## Author Contributions

**Conceptualization:** Elouise E. Kroon, Allison Seeger, Marianna Orlova, Marlo Möller, Robert J. Wilkinson, Anna K. Coussens, Erwin Schurr, Eileen G. Hoal.

**Data curation:** Elouise E. Kroon, Allison Seeger, Lize Engelbrecht, Jurgen A. Kriel, Ben Loos, Naomi Okugbeni, Marianna Orlova, Pauline Cassart, Craig J. Kinnear, Marlo Möller, Robert J. Wilkinson, Anna K. Coussens, Erwin Schurr, Eileen G. Hoal.

**Formal analysis:** Elouise E. Kroon, Wilian Correa-Macedo, Rachel Evans, Allison Seeger, Lize Engelbrecht, Jurgen A. Kriel, Ben Loos, Naomi Okugbeni, Marianna Orlova, Gerard C. Tromp, Marlo Möller, Robert J. Wilkinson, Anna K. Coussens, Erwin Schurr, Eileen G. Hoal.

**Funding acquisition:** Elouise E. Kroon, Marianna Orlova, Craig J. Kinnear, Marlo Möller, Robert J. Wilkinson, Anna K. Coussens, Erwin Schurr, Eileen G. Hoal.

**Investigation:** Elouise E. Kroon, Allison Seeger, Marlo Möller, Anna K. Coussens, Erwin Schurr, Eileen G. Hoal.

**Methodology:** Elouise E. Kroon, Rachel Evans, Allison Seeger, Lize Engelbrecht, Ben Loos, Gerard C. Tromp, Marlo Möller, Robert J. Wilkinson, Anna K. Coussens, Erwin Schurr, Eileen G. Hoal.

**Project administration:** Elouise E. Kroon, Marlo Möller.

**Resources:** Elouise E. Kroon, Marlo Möller, Robert J. Wilkinson, Anna K. Coussens, Erwin Schurr, Eileen G. Hoal.

**Supervision:** Gerard C. Tromp, Marlo Möller, Robert J. Wilkinson, Anna K. Coussens, Erwin Schurr, Eileen G. Hoal.

**Validation:** Elouise E. Kroon, Erwin Schurr.

**Visualization:** Elouise E. Kroon, Allison Seeger, Lize Engelbrecht, Jurgen A. Kriel, Anna K. Coussens, Erwin Schurr, Eileen G. Hoal.

**Writing – original draft:** Elouise E. Kroon.

**Writing – review & editing:** Wilian Correa-Macedo, Rachel Evans, Allison Seeger, Lize Engelbrecht, Jurgen A. Kriel, Ben Loos, Naomi Okugbeni, Marianna Orlova, Pauline Cassart, Craig J. Kinnear, Gerard C. Tromp, Marlo Möller, Robert J. Wilkinson, Anna K. Coussens, Erwin Schurr, Eileen G. Hoal.

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
