## [Decision Letter · Decision Letter 0]

13 Jun 2023

Dear Dr Kroon,

Thank you very much for submitting your Research Article entitled 'Altered neutrophil extracellular traps in response to Mycobacterium tuberculosis in persons living with HIV with no previous TB and negative TST and IGRA' to PLOS Genetics.

The manuscript was fully evaluated at the editorial level and by independent peer reviewers. The reviewers appreciated the attention to an important problem, but raised some substantial concerns about the current manuscript. Based on the reviews, we will not be able to accept this version of the manuscript, but we would be willing to review a much-revised version. We cannot, of course, promise publication at that time.

If you decide to revise the manuscript for further consideration at PLOS Genetics, please aim to resubmit within the next 60 days, unless it will take extra time to address the concerns of the reviewers, in which case we would appreciate an expected resubmission date by email to plosgenetics@plos.org.

We are sorry that we cannot be more positive about your manuscript at this stage. Please do not hesitate to contact us if you have any concerns or questions.

Yours sincerely,

Cathy Stein

Guest Editor

PLOS Genetics

Scott Williams

Section Editor

PLOS Genetics

The reviewers generally had a positive assessment of this work. However, there are many points that require clarification.

Reviewer's Responses to Questions

**Comments to the Authors:**

Reviewer #1: In their manuscript "Altered neutrophil extracellular traps in response to Mycobacterium tuberculosis in persons living with HIV with no previous TB and negative TST and IGRA", Kroon and colleagues explore neutrophil gene programs that distinguish a clinical phenotype of great interest. They compare well-defined cohorts of PLWH in South Africa who have presumed exposure to M. tuberculosis, which is based both on residence in a high endemic setting and Mtb-specific antibody responses, in order to explore potential mechanisms by which some immune-reconstituted PLWH lack canonical Mtb sensitization (TST/IGRA reactivity) with the hope to identify non-canonical mechanisms that protect them from TB progression. The manuscript is well crafted and written, and while it builds on a growing literature that explores IFNg independent mechanisms of TB protection among similarly defined clinical phenotypes (TST/IGRA based), the authors add novelty by focusing on PLWH where protective mechanisms may be distinct. Another strength is the inclusion of immunofluorescent microscopy evidence of NET formation to validate transcriptional findings.

While overall the composition is clear, there are a few areas where the narrative that builds from global transcriptional profiling (e.g. DEG identification and pathway enrichments) to experimental NET data could be made more clear.

Major comments:

Title:

- Title is confusing, perhaps because TST and IGRA syntax is awkward. Consider an alternative such as "Neutrophil extracellular trap formation and gene programs distinguish TST/IGRA sensitization outcomes among M. tuberculosis exposed persons living with HIV."

Figures 2 and 3 and corresponding section (lines 189-210):

The various analyses lead to some confusion assuming the overall conclusion of this section appears in the section heading. Examples and suggestions:

1) Figure 2 - Can a statistical statement be made whether the 6hour distributions (HITTIN versus HIT) are different?

2) The importance of the log2FC correlations (Fig 3) is not evident and leads to confusion based on the section heading.

- the statement (line 199) "differential expression across phenotypes was correlated with log2FC" needs to be better defined. To what the 'differential expression' refer since log2FC also describes differential expression (media - infection).

- Am I correct in stating the null hypothesis in this sense has not been rejected (e.g. phenotype does not impact the global Mtb PMN response)? If that is true, I would consider moving Fig3 to supplemental and make a simple statement in Results such as 'despite overall lower log2FC among HITTIN, there was a strong correlation of log2FC values for each gene between HITTIN and HIT suggesting that Mtb responses globally are conserved across phenotypes.'

3) Suggest removing 1h data in main results section to further streamline main conclusion and avoid unnecessary comparisons. It seems authors have concluded 1h is too early given insufficient transcriptional changes. The statement justifying a focus on 6h time point could be make earlier with reference to supplemental info for 1h timepoint data).

HITTIN vs HIT interaction DEG model:

(lines 205 - 210) - The model used when comparing HITTIN against HIT needs [a brief] reference in main results section. Presumably these 2285 DEGs reflect the interaction between (uninfected - Mtb) *(HIT - HITTIN), but is restricted to the 6h timepoint? As written, it suggests a simple contrast model was used that is restricted to Mtb-infected samples (6h) only. I also note in the methods section that infection time (1h - 6h) was also included in the DEG model that may further lend confusion to interpretation of this main results section (and 2285 DEGs).

Directionality of NET term enrichments (and NADPH oxidase DEGs).

(line 229) Reference to 'terms that directly relate to a possible increased microbicidal activity of PMN-HITTIN' leads to a confusing directionality based on subsequent sections that highlight genes that actually have reduced expression in HITTIN.

- (line 230) should be 'Fig 4B'.

- NADPH oxidase genes (RAC2, CYBA, CYBB, NFKB1, NCF2) had decreased expression among HITTIN as described in Results and negative log2FC in Table 3. The positive enrichment among HITTIN of the NET formation pathway (higher NETs among HITTIN?) lends well to the later experimental data but seems contradictory to the gene-level directionality that is highlighted.

- Reordering these sections by including the NET microscopy data immediately after reporting the positive HITTIN enrichment for the NET formation pathway in 4B may be easier to follow. Once the conclusion that NETs are lower in HITTIN is made, then further exploration at the gene-level can be made (Table 3 and results section 'DEGs in HITTIN vs HIT...').

Limitations

- Considering HITTIN have lower magnitude of transcriptional changes following Mtb infection overall, a contribution from non-PMN populations in the experiment should be considered. Contaminating T cells could plausibly contribute to IFNg-driven PMN transcriptional differences that are detected according to the pre-defined clinical phenotype.

- It would be appropriate to mention this limitation, and that good PMN purity (90%, T cells ~5%) and the 6h timepoint that is short for paracrine effects argue against a significant impact of contaminating cell types.

- Considering the sensitivity of RNAseq it would be further reassuring to confirm there are not other IFNg pathway-related enrichments between HIT vs. HITTIN.

- Note that IFNg even after short stimulation times can impact PMN NADPH oxidase expression (and NADPH production) in vivo:

- https://doi.org/10.1371/journal.pone.0263370

- https://doi.org/10.1182/bloodadvances.2021005776

Minor comments:

- Data access: Submission of data to the European Genome-phenome Archive (EGA) is currently being reviewed. Please confirm submission is processed/finalized.

- (lines 59-61; awkward syntax); Consider "some PLWH never develop TB and show no evidence of immune sensitization to Mycobacterium tuberculosis (Mtb) as defined by persistently negative tuberculin skin tests (TST) and interferon gamma release assays (IGRA)."

- Consider including FDR threshold in Abstract to define significant

- (lines 67 - 72) - differentially expressed is not defined initially and leads to confusion since the primary comparison as defined above is HITTIN versus HIT. Suggest better delineating with "When compared to uninfected PMNs, PMNhittin displayed 151 unregulated and 40 down regulated differentially expressed genes (DEGs) (FDR XX) whereas PMNhit demonstrated 98 upregulated and 11 downregulated DEGs following 1h of Mtb infection.

- (line 72) - As above, consider adding "...3794 significantly downregulated DEGs when comparing Mtb-infected and uninfected PMNs.

- (lines 111 - 120) Is there a reason why the PMN abbreviation is not assigned after first neutrophil use, or at least why this sentence is chosen for PMN abbreviation when neutrophil is spelled out later as well?

- (line 155-158) consider change to "...11 HIT individuals, all of whom were PLWH and on ART, were used in the final analysis (Table 1). These individuals were part of stringently defined cohorts living in a high TB burden community who despite low CD4+ counts before ART initiation never developed TB."

- Figure 1 - Legend suggests vertical line reflects log2FC of -/+ 0.02 whereas results section (and visual inspection) suggests this should be -/+ 0.2.

- Figure 4 - legend (line 1057) should have 'HIT' not 'HT'.

- (line 230) reference should be Fig 4B.

- (line 244-245) change to "NADPH oxidation by Nox2 (encoded by CYBB) and other cellular NADPH oxidases is involved with ROS production and NET formation (29-31)."

- (line 277) should be "induced at 1h and 6h post infection".

- (lines 291 - 292) Incomplete sentence.

- (line 322) change to "Mtb-induced ROS triggered necrosis in neutrophils..."

- (line 349) change to "TLR2/4 signaling could also mediate NET formation independent of ROS" or "ROS-independent mechanisms could also mediate NET formation downstream of TLR2/4."

- (line 371-372) suggest change to "....maintained by PMN-HITTIN, which demonstrate lower NET formation in response to Mtb infection despite a positive enrichment of NET-related genes as compared to PMN-HIT."

Reviewer #2: In this manuscript, Kroon and colleagues present the results of a RNAseq study on neutrophils in response to Mycobacterium tuberculosis (Mtb) challenge in persons living with HIV who are at high risk of TB. They compared the transcriptome profile of neutrophils after Mtb infection in 17 HIV+ persistently TB, tuberculin and IGRA negative (HITTIN) participants living in a community with high TB burden and 11 individuals living with HIV from the same community with no TB history, but who test persistently IGRA positive, and tuberculin positive (HIT). After 6h of infection, neutrophils of HITTIN participants showed an overall transcriptional impairment as compared to HIT participants, consistent with the lower number of differentially expressed genes (DEGs) in HITTIN vs HIT sujects (3106 vs 3816 up-regulated genes and 3548 vs 3794 down-regulated genes). When comparing the response to Mtb between HITTIN and HIT individuals, the authors identified 2285 significant genes. Genes with a significant positive fold change (less downregulated by neutrophils from HITTIN individuals, N=1068) were enriched in « Apoptosis », « Neutrophil extracellular trap formation », and “NADPH regeneration” pathways. Interestingly, lower neutrophil extracellular trap formation was observed by fluorescence microscopy in HITTIN compared to HIT participants.

This manuscript tackles an important subject in the field of tuberculosis and suggest that NETosis could play a role in the early control of Mtb infection. I do, however, have some concerns regarding the robustness and interpretation of the findings.

1) My main concern pertains to the overall transcriptional impairment after 6h of Mtb infection observed in neutrophils of HITTIN participants compared to HIT participants. I am wondering whether it could be attributed to systematic differences not accounted for between the two groups that influence the transcriptional responsiveness of neutrophils to Mtb. Notably, previous studies by some of the authors have demonstrated that antiretroviral therapy (ART) significantly affects the transcriptional responsiveness of alveolar macrophages to Mtb (Correa-Macedo et al., JCI, 2021). Hence, I am curious if the duration of ART treatment in the participants could partially account for the observed overall impairment in neutrophils. It is essential to investigate, account for, and discuss this possibility.

2) Related to the previous point, I am also concerned about how this transcriptional impairment may have impacted the results of the differential gene expression (DEG) analysis between HITTIN and HIT individuals. The ability to detect differences between the two groups is likely higher for genes that exhibit a robust response in HIT participants following Mtb infection. Consequently, I am wondering whether the enriched pathways could reflect the general response of neutrophils to Mtb rather than a differential expression pattern between HITTIN and HIT groups. It would be helpful to show the results of the pathway enrichment analysis in response to Mtb performed separately for HITTIN and HIT. Are the terms « Apoptosis », « Neutrophil extracellular trap formation », and “NADPH regeneration” significantly enriched in differentially expressed genes (especially down-regulated)? How do they rank in terms of significance in the two groups? This should be discussed.

3) I am wondering if normalizing the expression profile of the HITTIN subject after 6 hours of Mtb infection to render it more comparable to HIT participants could make sense and how it could impact on the results of the DEG analysis between HITTIN and HIT groups?

Reviewer #3: Very well written paper. This explores an important and relevant topic (understanding the immune response to TB infection in HITTIN and more broadly PLWH) and the authors explain and lay out the justification for this research appropriately. The introduction appropriately lays out the relevant background information on immune responses to TB, and the role of neutrophils/PMN's. Methodologically, the HIT group is an appropriate comparator for this study and the selection of subjects was adequately explored and justifiable, including an exploration of risk factors which would potentially serve as confounders. Differential gene expression in neutrophils/PMN's is an appropriate outcome for this question and the analysis was performed according to accepted standards. The authors further added to their understanding of the neutrophil response by examining the functional role of the DEG's they identified (showing enrichment for extracellular traps). Their microscopy validation was a nice way to wrap this up.

Overall, this answers an important research question and the study is appropriately performed and the paper is well written. The results are impactful and significant. The discussion was also well-written and helps contextualize the importance of these results.

**Have all data underlying the figures and results presented in the manuscript been provided?**

Reviewer #1: Yes

Reviewer #2: Yes

Reviewer #3: Yes

PLOS authors have the option to publish the peer review history of their article (what does this mean?). If published, this will include your full peer review and any attached files.

Reviewer #1: No

Reviewer #2: No

Reviewer #3: No

---

## [Decision Letter · Decision Letter 1]

26 Jul 2023

Dear Dr Kroon,

We are pleased to inform you that your manuscript entitled "Neutrophil extracellular trap formation and gene programs distinguish TST/IGRA sensitization outcomes among Mycobacterium tuberculosis exposed persons living with HIV" has been editorially accepted for publication in PLOS Genetics. Congratulations!

Yours sincerely,

Cathy Stein

Guest Editor

PLOS Genetics

Scott Williams

Section Editor

PLOS Genetics

Comments from the reviewers (if applicable):

You may want consider making the changes suggested by the Reviewer 1 regarding wording, as these will strengthen the paper.

Reviewer's Responses to Questions

**Comments to the Authors:**

Reviewer #1: All previous recommendations have been appropriately addressed. Several minor notes/considerations are as follows should authors/editor choose to modify.

Abstract

- penultimate sentence -- Suggest to avoid 'enrichment of genes in PMN-HITTIN' that may erroneously be understood by the reader as a pathway that is transcriptionally unregulated. An alternative could be "According to pathway enrichment, Aptotosis and NETosis were differentially regulated between HITTIN and HIT PMN responses after 6h Mtb infection. To corroborate the blunted NETosis transcriptional response measured among HITTIN, fluorescence microscopy revealed relatively lower neutrophil extracellular trap formation and cell loss in PMN-HITTIN as compared to PMN-HIT.'

- Would suggest a broad conclusion final sentence after (maybe with an impact or future study statement).

- A major (?) conclusion that global transcriptional responses are lower in HITTIN is missing from abstract (yet appears in author summary) whereas very specific data from 1h time point is included. Considering 1h information was moved to supplemental, consider revisiting this choice for abstract content.

Author summary - should be "..we examined neutrophil responses to Mtb and compared HITTIN to HIT" or otherwise clarify since compare refers to two different 'to'.

Minor issues

(4th line abstract) - delete extra period

- Sentence "We used fluorescent microscopy to evaluate the biological outcome..." seems to better fit under the heading that follows this paragraph. It would benefit from a transition to link findings from previous section, maybe just by reordering "To directly evaluate the biological outcome in the total amount of NETs observed between HITTIN vs. HIT, as is suggested by our transcriptional enrichments, we used fluorescent microscopy to quantitate NET formation." Also, the reference to Table 3 in this original sentence seems out of place (refers to following DEG heading, which already has this reference).

- Limitations - should be "we employed subject-specific fold changes."

Reviewer #2: I would like to thank the authors for their time and effort to answer my comments. I have no additional concern.

**Have all data underlying the figures and results presented in the manuscript been provided?**

Reviewer #1: Yes

Reviewer #2: Yes

PLOS authors have the option to publish the peer review history of their article (what does this mean?). If published, this will include your full peer review and any attached files.

Reviewer #1: No

Reviewer #2: No

**Data Deposition**

http://datadryad.org/submit?journalID=pgenetics&manu=PGENETICS-D-23-00472R1

**Press Queries**

---

## [Editor Report · Acceptance letter]

18 Aug 2023

PGENETICS-D-23-00472R1 

Neutrophil extracellular trap formation and gene programs distinguish TST/IGRA sensitization outcomes among *Mycobacterium tuberculosis* exposed persons living with HIV 

Dear Dr Kroon, 

We are pleased to inform you that your manuscript entitled "Neutrophil extracellular trap formation and gene programs distinguish TST/IGRA sensitization outcomes among *Mycobacterium tuberculosis* exposed persons living with HIV" has been formally accepted for publication in PLOS Genetics! Your manuscript is now with our production department and you will be notified of the publication date in due course.

With kind regards,

Jazmin Toth

PLOS Genetics

On behalf of:
